# The patatin-like protein PlpD forms structurally dynamic homodimers in the *Pseudomonas aeruginosa* outer membrane

Sarah E. Hanson[1], Tyrone Dowdy [2], Mioara Larion [2], Matthew Thomas Doyle [1,3] ✉ & Harris D. Bernstein [1] ✉

Members of the Omp85 superfamily of outer membrane proteins (OMPs) found in Gram-negative bacteria, mitochondria and chloroplasts are characterized by a distinctive 16-stranded β-barrel transmembrane domain and at least one periplasmic POTRA domain. All previously studied Omp85 proteins promote critical OMP assembly and/or protein translocation reactions. *Pseudomonas aeruginosa* PlpD is the prototype of an Omp85 protein family that contains an N-terminal patatin-like (PL) domain that is thought to be translocated across the OM by a C-terminal β-barrel domain. Challenging the current dogma, we find that the PlpD PL-domain resides exclusively in the periplasm and, unlike previously studied Omp85 proteins, PlpD forms a homodimer. Remarkably, the PL-domain contains a segment that exhibits unprecedented dynamicity by undergoing transient strand-swapping with the neighboring β-barrel domain. Our results show that the Omp85 superfamily is more structurally diverse than currently believed and suggest that the Omp85 scaffold was utilized during evolution to generate novel functions.

The proteins that reside in the outer membrane (OM) of Gram-negative bacteria perform a wide variety of functions, including the uptake of small molecules[1], the secretion of virulence factors[2,3] and the maintenance of the OM itself[4]. The last group both promotes the membrane insertion of new proteins and the maintenance of an asymmetric bilayer that contains phospholipids in the inner leaflet and a unique glycolipid called lipopolysaccharide (LPS) in the outer leaflet[5–7]. Outer membrane proteins (OMPs) are unusual in that they lack α-helical membrane spanning segments but instead span the membrane via a "β-barrel", which is essentially an amphipathic β-sheet (with a hydrophilic interior and hydrophobic exterior) folded into a closed cylindrical structure[8–10]. β-barrels are almost always monomeric or homotrimeric and are usually extremely stable due to a hydrogen bond network that holds the first and last strands together[8,10]. They are also highly diverse in sequence and contain from 8-36 β-strands[8,11,12].

Some OMPs consist solely of an empty β-barrel, while others are plugged and/or are linked to periplasmic or extracellular domains[8,13]. All OMPs are synthesized in the cytoplasm and transported across the inner membrane via the universal Sec machinery and then interact with periplasmic chaperones (e.g. SurA/Skp), which prevent them from misfolding[14]. OMPs are then targeted to the barrel assembly machinery (BAM) complex, an essential and highly conserved heterooligomer that catalyzes the folding and membrane insertion of OMPs[6,7]. The core subunit of the BAM complex, BamA, is itself an OMP that contains a 16-stranded β-barrel plus five polypeptide transport-associated (POTRA) domains that mediate the attachment of several accessory proteins in the periplasm[15–20]. Although the mechanism of the BAM complex is not well understood, available evidence indicates that the BamA β-barrel is unusually unstable[17,20,21]. This property allows its first and last strands to open laterally to perturb the local membrane structure and to

---

[1]Genetics and Biochemistry Branch, National Institutes of Diabetes and Digestive and Kidney Diseases, National Institutes of Health, Bethesda, MD 20892, USA. [2]Neuro-Oncology Branch, National Cancer Institute, National Institutes of Health, Bethesda, MD 20892, USA. [3]Sydney Infectious Diseases Institute and School of Medical Sciences, Faculty of Medicine and Health, The University of Sydney, Sydney, New South Wales 2006, Australia. ✉ e-mail: m.doyle@sydney.edu.au; harris_bernstein@nih.gov

directly associate with OMPs during the folding and insertion process[22,23].

BamA is a member of a diverse superfamily of OMPs known as the Omp85 proteins that are ubiquitous in Gram-negative bacteria and organelles of bacterial origin (i.e., mitochondria and chloroplasts). The proteins in this superfamily contain a distinctive C-terminal 16-stranded β-barrel and usually one or more periplasmic POTRA domains, and often carry out functions that are essential for the maintenance of the cell[24,25]. In some cases, POTRA domains are involved in the binding of accessory proteins or the binding or release of substrates, but in other cases, their function remains unknown[19,26–30]. Although 10 families of Omp85 proteins have been identified, all of the members that have been characterized to date catalyze the assembly of other OMPs or the translocation of proteins across the OM[24,25]. In addition to BamA and its functional equivalent in mitochondria (Sam50)[31], TamA promotes the assembly of a subset of OMPs in concert with TamB, an inner membrane protein that spans the periplasm[32,33]. It should be noted that TamB has recently been shown to promote the trafficking of lipids between the two membranes in *E. coli*, and TamA might also play a role in this process[34,35]. The TpsB family proteins are a component of the so-called two partner secretion (TPS or type Vb secretion) systems that are dedicated to the translocation of a single co-transcribed "exoprotein" (TpsA) across the bacterial OM, and Toc75 (together with other components of the TOC complex in the OM and the TIC complex in the inner membrane) translocates proteins from the cytoplasm into the stroma of chloroplasts[36–40]. Other members of the Omp85 superfamily, however, have not been well characterized, including several highly conserved subfamilies that contain N-terminal metalloprotease, WD40-like, or patatin-like domains[25]. These Omp85 proteins might have very different functions and are, therefore, intriguing targets for structural and functional studies.

Sequence analysis suggests that the patatin-like (PL-Omp85) proteins, represented by the archetypical *P. aeruginosa* protein patatin-like protein D (PlpD), are composed of the characteristic Omp85 16-stranded β-barrel, one POTRA domain, and an N-terminal PL-domain connected to the POTRA via a short linker[25,41]. The PL-domain is similar to the potato-derived patatin protein, a phospholipase characterized by an α/β hydrolase fold and a Ser-Asp catalytic dyad, as well as four conserved sequence motifs[41,42]. Recombinant forms of the PL-domain and its homologs have been shown to have phospholipase activity in vitro[42,43]. Based on the observation of a PlpD fragment located in the supernatant of *P. aeruginosa* cultures, it was proposed that the PL-domain is translocated across the OM through the β-barrel and then released by proteolytic cleavage into the extracellular space, where it might act as a virulence factor[41]. This model of biogenesis is very similar to that of the type Vb TPS system, but differs in that the extracellular component (i.e. the PL-domain) is covalently linked to the Omp85 transporter component[42]. The apparent similarities to the TPS and other type V secretion systems that share conceptual similarities in their proposed biogenesis pathways led to the designation of PL-Omp85 proteins as a novel "type Vd" secretion system[41]. In more recent studies, however, it was unclear if the *Fusobacterium nucleatum* PL-Omp85 protein FplA is secreted[44]. To add to the uncertainty, the finding that the PlpD PL-domain crystallized as a dimer raised questions about whether dimerization occurs before or after secretion and if the protein truly associates as a dimer under physiological conditions[42]. Overall, the topology and biogenesis of PL-Omp85 proteins and their role in cell physiology remain a mystery.

Here we sought to obtain insight into the structure of PlpD in vivo in a virulent clinical isolate of *P. aeruginosa* (strain PA14[45]). To this end, we utilized proteases to probe for surface-exposed domains and cysteine crosslinking based on structural models predicted by the AI software AlphaFold to evaluate the proximity of various domains to each other and to neighboring PlpD molecules. We obtained clear evidence that the PL-domain of PlpD remains in the periplasm but observed an intracellular breakdown product that might be present in the culture medium at a low level under certain laboratory conditions. Our localization of the PL-domain is not only supported by protease sensitivity and domain proximity data, but also by functional data that showed that the deletion of *plpD* strongly increases the level of membrane lipids that would be accessible to the phospholipase domain only if it were located in the periplasm. Additionally, we find that the entire PlpD protein—not just the PL-domain—dimerizes in vivo and represents the first dimer in the Omp85 superfamily (and one of the few known dimeric OMPs). Lastly, we found that an unsolved region of the PL-domain crystal structure (the lipid binding pocket "lid") has unprecedented dynamicity and interacts with the first and last strands of both PlpD β-barrel domains. Our findings provide important clues about the possible function of the highly unusual and widespread patatin-like Omp85 protein family.

## Results
### The PlpD patatin-like domain is located in the periplasm

Because PlpD is produced at a low level and is difficult to detect when *P. aeruginosa* is grown under laboratory conditions (Fig. S1), we cloned *plpD* into the broad host range plasmid pSCrhaB2 under the control of a rhamnose-inducible promoter[46] to determine if the PL-domain is translocated across the OM as previously proposed[41]. To aid detection of the protein, we added a TwinStrepII (TS) tag at the N-terminus of the PL-domain (Fig. 1a). Like most OMPs that are inserted into the OM and correctly folded, we found that both tagged and untagged versions of the protein are tightly folded and resistant to SDS denaturation unless they are heated (Fig. S2). *P. aeruginosa* strain PA14 Δ*plpD* (SEH88) transformed with the plasmid was grown in LB at 37°C. PlpD expression was induced by the addition of 0.2% rhamnose, and culture samples were subsequently collected and divided into whole cell and supernatant fractions.

In contrast to previously reported results[41], we detected a band corresponding to full-length PlpD (~82 kDa) in cell pellet samples by Western immunoblotting with either an anti-TS tag (αStrepII) antibody or an antiserum generated against a C-terminal PlpD peptide (αPlpD$_C$), but we did not detect a fragment that corresponds to the PL-domain in the supernatant (Fig. 1b). These results strongly suggest that the PL-domain is not released from cells under our experimental conditions. Interestingly, we observed a ~36 kDa N-terminal band that corresponds to an N-terminal PlpD degradation product in the cell pellets. Although larger ~45 kDa bands were previously observed in culture supernatants[41] (a molecular weight that corresponds to an N-terminal fragment containing both the PL- and POTRA domains), it seems possible that those bands represent a periplasmic degradation product that was found in the supernatant due to cell lysis. It is also possible, however, that the PL-domain is secreted across the OM as a "passenger" domain but remains associated with the cell surface under our experimental conditions. To test this possibility, we expressed PlpD and permeabilized the OM of half of the cells before treating all the samples with proteinase K (PK; Fig. 1c). If the PL-domain is exposed on the extracellular side of the OM, it is likely that we would see its release by PK even when the OM remains intact. We found that PK cleaved the protein even without OM permeabilization, but curiously, protease treatment did not release the PL-domain. Instead, the protease generated two fragments in roughly equal abundance, a large fragment that presumably represents full-length PlpD with the TS-tag removed, and a ~38 kDa C-terminal fragment that corresponds to the β-barrel domain and at least a part of the POTRA domain, a domain that would be expected to reside in the periplasm. The data suggest that the full-length protein (but not the TS-tag) is folded and is mostly refractory to digestion, but that the protease had access to the POTRA domain. Essentially the same results were obtained when cells that produced a tagless version of PlpD were treated with PK, except that as expected the PK treatment did not alter the size of the large

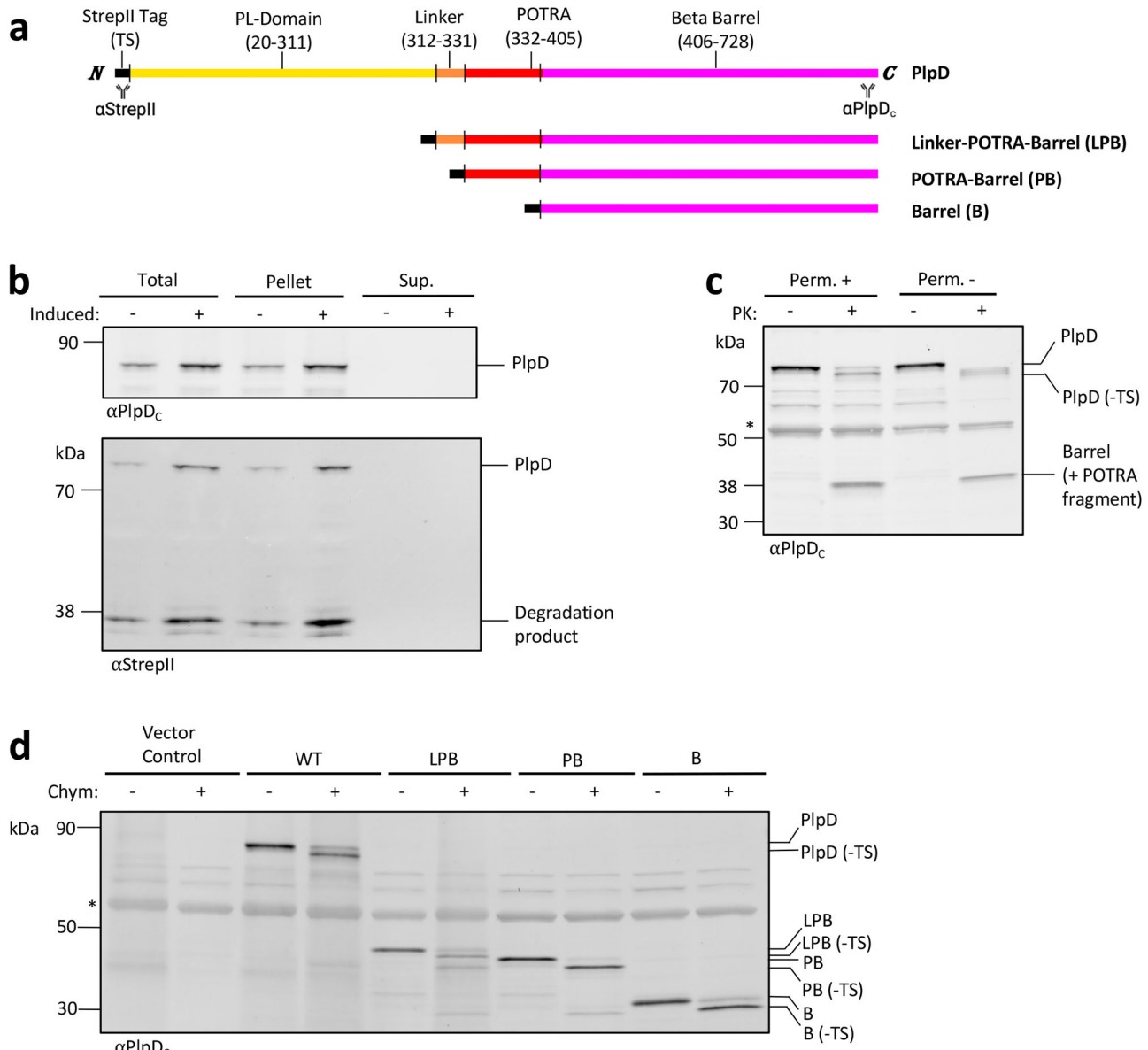

**Fig. 1 | The PlpD PL-domain is not secreted. a** Primary structures of the PlpD constructs used in this study. The location of protein segments [β-barrel (magenta), POTRA domain (red), linker (orange), PL-domain (yellow) and StrepII tag (black)] are shown. Mouse-anti-StrepII and rabbit-anti-PlpD$_C$ antibody binding sites are indicated. **b** *P. aeruginosa* strain SEH88 transformed with pSEH81 were grown to mid-log phase and the culture was divided in half. The expression of *plpD* was induced in one half by the addition of 0.2% L-rhamnose as described in Methods. Proteins in the total culture (Total), cell pellet (Pellet), and supernatant (Sup.) were separated by SDS-PAGE. Immunoblots were then conducted using the indicated antisera. **c** SEH88 transformed with pSEH81 were grown and the expression of *plpD* was induced as in (b). After the cells were pelleted, resuspended in PBS, and divided in half, the OM of one half was permeabilized (Perm. + ) while the OM of the other half remained intact (Perm. -). Aliquots were then either treated with PK or left untreated. Samples were analyzed by immunoblot using the anti-PlpD$_C$ antiserum. The asterisk denotes a non-specific cross-reactive band. **d** SEH88 transformed with the empty pRha vector, pSEH81, pSEH93, pSEH95, or pSEH96 were grown and the expression of wild-type *plpD* (WT) or the indicated derivative was induced as in (b). Subsequently cells were treated with chymotrypsin or mock-treated. Immunoblots were then conducted using the anti-PlpD$_C$ antiserum. The asterisk denotes a non-specific cross-reactive band. The experiments in (b-d) were performed twice with similar results.

polypeptide (Fig. S3a). The results of the protease treatment experiments thereby corroborate the conclusion that the PL-domain resides in the periplasm but is sensitive to cleavage by endogenous proteases.

To further analyze the topology of PlpD, we next compared the protease sensitivity of PlpD with that of truncated derivatives that contain 1) an N-terminal TS-tag fused to the linker, POTRA domain and β-barrel (LPB), 2) the POTRA domain and β-barrel (PB), or 3) only the β-barrel (B) (Fig. 1a). Each PlpD derivative was expressed in SEH88 and half of the cells were treated with chymotrypsin, a protease that cleaves at aromatic residues. Because aromatic residues are usually shielded within the hydrophobic core of a folded protein domain or by

OM lipids in vivo, chymotrypsin treatment can yield information about protein structure. In intact cells that expressed full-length PlpD only the TS-tag was susceptible to chymotrypsin digestion, and in cells that expressed full-length untagged PlpD the protein was completely resistant to the protease (Fig. 1d and S3b). However, multiple bands corresponding to C-terminal fragments containing a portion of the POTRA domain plus the β-barrel or only the β-barrel were observed in cells that expressed either the LPB or PB constructs. The results indicate that aromatic residues within the POTRA domain or near the interface between the POTRA and β-barrel domains were exposed to the protease in the absence of the PL-domain. While it is unlikely that

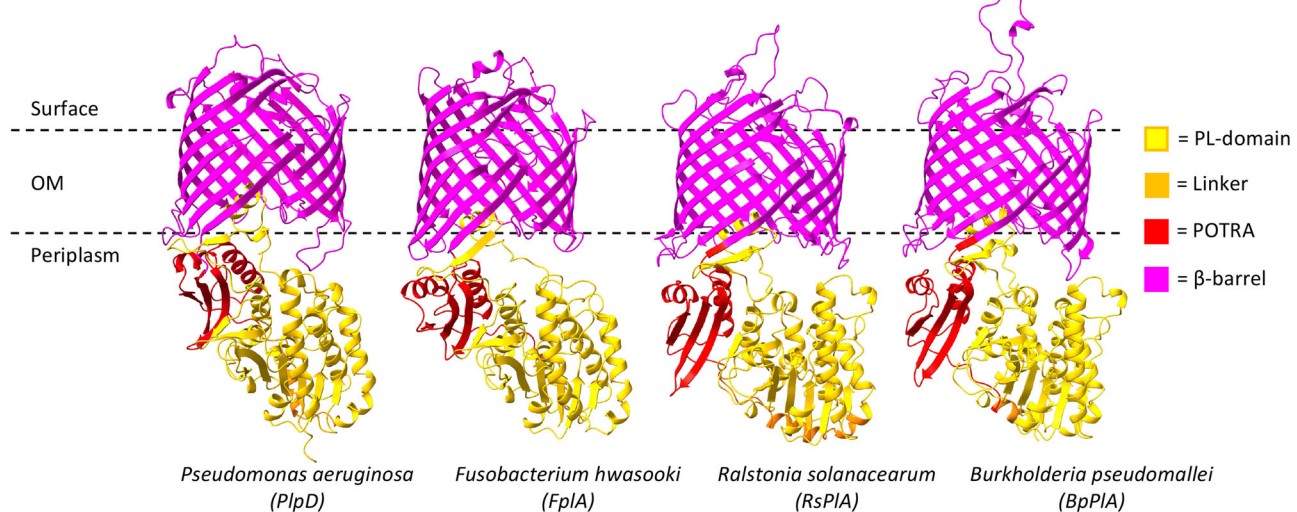

**Fig. 2 | Structure predictions indicate that the conformation of the PL-Omp85 family is conserved.** AlphaFold models of PlpD (UniParc: UPI00054992B6) and three of its homologs, *Fusobacterium hwasooki* FplA (UniParc: UPI00027C9D86; 23% identical and 42% similar to PlpD), *Ralstonia solanacearum* RsPlA (UniParc: UPI000066CB11; 34% identical and 47% similar to PlpD) and *Burkholderia pseudomallei* phospholipase A (BpPlA; UniParc: UPI0005C3BCB; 34% identical and 49% similar to PlpD) are shown. Domains are color-coded as in Fig. 1.

the POTRA domain was exposed on the cell surface given that POTRA domains of other Omp85 family members have never been observed outside of the periplasm, the cleavage of the N-terminal TS-tag from the β-barrel-only derivative is particularly striking because it is predicted to be too short to pass through the lumen of the barrel to the extracellular side of the OM. The most likely explanation of the data is that the OM of *P. aeruginosa* is particularly sensitive to permeabilization by proteases and that the unstructured TS-tag and unshielded aromatic residues of truncated PlpD constructs are cleaved on the periplasmic side of the membrane. Consistent with this interpretation, we found that a large fraction of the periplasmic chaperone SurA was also digested by proteases when the membrane was not permeabilized (Fig. S4). Furthermore, we found that when PlpD or PlpD fragments are expressed in *E. coli*, an organism whose OM remains intact after PK treatment, they fold and insert into the OM properly[22], but the POTRA domain and TS-tag are cleaved only when the membrane is first permeabilized (Figs. S5, S6).

Given that the PL-domain does not appear to be translocated across the *P. aeruginosa* OM, we next sought to determine the position of this domain relative to other segments of PlpD. Coincidentally, tertiary structure predictions of all *P. aeruginosa* open reading frames were deposited into the AlphaFold Database while we were analyzing the topology of PlpD[47]. Strikingly, AlphaFold predicted that the PL-domain is not only located in the periplasm (Fig. 2, yellow), but that it forms an interface with the POTRA domain (Fig. 2, red) in alignment with our results. The linking sequence between the PL- and POTRA domains is predicted to form an α-helix that is also exposed to the periplasmic milieu (Fig. 2, orange). Almost the entire structure of the protein was predicted with high confidence (Fig. S7). Furthermore, AlphaFold predicted that even distantly related PL-Omp85 homologs fold into essentially the same conformation (Fig. 2).

To test the predicted structure experimentally, we identified residues in the PL- and POTRA domains that appear to be in close proximity and mutated them individually or in a pairwise fashion to cysteine, an amino acid that is absent in the native PlpD protein (Fig. 3a). We then expressed the cysteine mutants in SEH88 and incubated the cells with the oxidizing reagent 4,4′-dipyridyl disulfide (4-DPS) to determine if the cysteine Cβ atoms are close enough (within 3−5 Å of each other) to form disulfide bonds in vivo. Following the

addition of 4-DPS, samples were analyzed by immunoblots using the αPlpP$_C$ antiserum. As expected, none of the single cysteine mutants formed disulfide bonds (Fig. 3b). In contrast, we observed bands corresponding to PlpD, whose electrophoretic migration was retarded, indicating the presence of intramolecular disulfide bonds between N136C and Q383C, Q135C and Q383C, L145C and V388C, and V146C and V388C. It should be noted that intramolecular disulfide bonding has been observed to reduce the mobility of proteins previously[48], and the finding that the slow migrating bands disappeared when samples were treated with the reducing agent DTT confirmed that they arose from the formation of disulfide bonds between the introduced cysteines (Fig. S8). In the case of cysteine pairs that are in the most favorable predicted orientations (Q135C/Q383C and V146C/V388C), disulfide bonding levels were extremely high (83 ± 6% and 95 ± 2%, respectively) (Fig. 3c). This finding strongly suggests that the selected residue pairs are in very close proximity in the native state of PlpD in the OM. Taken together with the observation that the POTRA domains of all previously characterized Omp85 proteins are located in the periplasm, the intramolecular disulfide bonding data and AlphaFold predictions imply that PL-domains of the PL-Omp85 family are not secreted but instead remain in the periplasm in close association with the POTRA domain.

## PlpD is a homodimer

One limitation of the AlphaFold database is that it only provides structure predictions for single protein chains. The crystal structure of the PlpD PL-domain, however, was previously solved as a homodimer[42]. For this reason, we subsequently used AlphaFold Multimer[49] to determine whether the full-length PlpD protein might dimerize. Interestingly, the software predicts that PlpD forms a homodimer with high confidence scores (Fig. S7). The validity of the prediction is supported by the fact that the predicted structure of the PL-domain closely matches the solved crystal structure (Fig. S9). Intriguingly, the β-barrel domains are predicted to interact near an interface created by surface loops 1, 2, and 4 (Fig. 4a). Such an interaction is predicted to tilt the two barrel domains towards each other in a way that requires the displacement of LPS near the interface (Fig. S10). Indeed, the Orientations of Proteins in Membranes (OPM) server[50] predicts that PlpD would significantly bend the outer membrane plane (Fig. 4a).

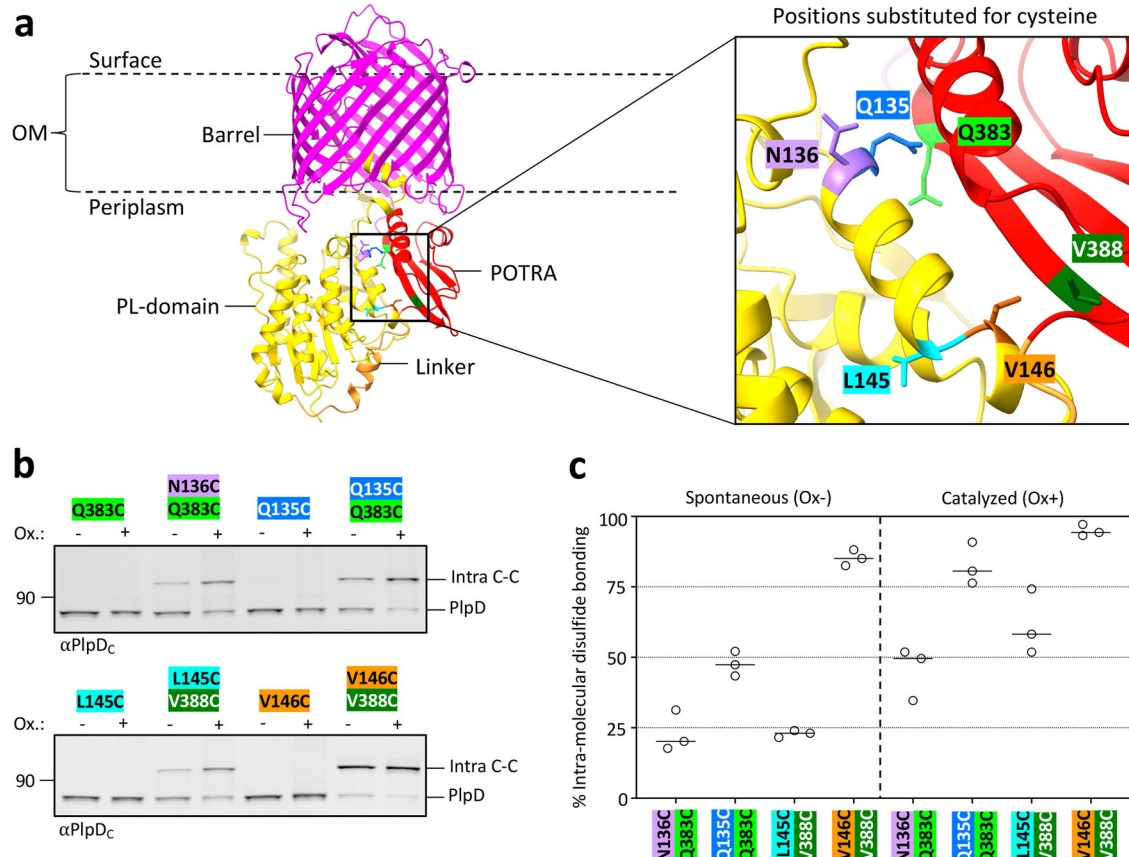

**Fig. 3 | The PL-domain of PlpD associates with the POTRA domain in vivo.**
**a** AlphaFold model of PlpD monomer (UniParc: UPI00054992B6) color-coded as in Fig. 1a. A portion of the POTRA and PL-domains is enlarged in the box on the right to highlight the relative location and orientation of the selected residues that were mutated to cysteine [PL-domain residues Q135 (blue), N136 (purple), L145 (cyan) and V146 (orange), and POTRA domain residues Q383 (light green) and V388 (dark green)]. **b** SEH88 transformed with pSEH171, pSEH172, pSEH202, pSEH203, pSEH211, pSEH238, pSEH264 or pSEH266 were grown and the expression of the indicated *plpD* cysteine mutants was induced as described in Fig. 1b. Aliquots were then either mock-treated or treated with the oxidant 4-DPS and samples were analyzed by immunoblot using the anti-PlpD$_C$ antiserum. Both the native

uncrosslinked PlpD protein (~82 kDa) and an intra-molecularly crosslinked form (~100 kDa) were detected. The ~100 kDa bands were confirmed to result from disulfide bond formation by adding DTT to the samples prior to SDS-PAGE (see Fig. S8). The colors of the residue labels correspond to those in (**a**). **c** Quantitation of intra-molecular disulfide bond formation between cysteine residues located in the PL-domain and POTRA domain. The levels of disulfide bond formation in mock-treated cells ('spontaneous', Ox-) and 4-DPS-treated cells ('catalyzed', Ox + ) are shown. Bars = median, *N* = 3 biologically independent samples that each contained cells grown from a different colony. One-sided ANOVA and multiple comparison tests are shown in Table S4.

To test the predicted dimeric structure in vivo we again used the disulfide bond analysis in SEH88 described above. Based on their orientation and location along the predicted dimeric interface, we chose to replace PL-domain residue M249 and β-barrel domain residues F427 (strand 2) and I532 (loop 4) with cysteine (enlarged in Fig. 4b). Position L359 served as a control as it was predicted to be distal to the dimerization interface. Consistent with the AlphaFold prediction, we found that about 65% of expressed PlpD molecules containing either the M249C or I532C substitution formed high molecular weight bands that correspond to intermolecularly cross-linked dimers after a 30 min incubation with 4-DPS, but that only a small fraction of the L359C mutant formed disulfide bonds (Fig. 4c, d). Similar to previously reported results[22,51–53], the formation of intermolecular crosslinks between different residues of PlpD slightly altered the mobility of the crosslinking products. Essentially the same intermolecular crosslinking pattern was observed when the experiment was repeated with an untagged version of PlpD (Fig. S11). Surprisingly, the F427C mutant formed intermolecular disulfide bonds at about the same level as the L359C mutant. It is possible that F427 is oriented inside the barrel lumen but that AlphaFold incorrectly predicted it as an outward facing residue. It should be noted, however, that the Hidden Markov Model (HMM) logo for PL-Omp85

proteins (Pfam ID: Omp85_2) shows that F427 is a conserved residue[54]. This finding suggests that F427 may be important for dimerization and that its mutation to cysteine might cause a slight conformational change that hinders the formation of intermolecular crosslinks. We also conducted oxidation time courses using cells that expressed either the M249C or I532C mutant. The inter-molecular disulfide bonding of the I532C mutant peaked at high levels (63%) almost immediately after the addition of 4-DPS (Fig. 4e and S12a). This observation suggests that loop 4 interacts stably with the cognate region of the protein in the PlpD dimer in the *P. aeruginosa* OM. In contrast, disulfide bonds between cognate M249C residues accumulated more slowly but likewise plateaued at a high level (91.5%) between 60-90 mins after the addition of the oxidizer (Fig. 4f and S12b). Presumably the relatively slow accumulation of disulfide bonds is due to a positioning of the M249C mutant within the hydrophobic PL-domain dimer interface that shields the residue from the oxidizing agent or, as indicated by the PL-domain crystal structure, the cognate residues are oriented in opposite directions[42]. Finally, the finding that high molecular weight bands were observed on SDS-PAGE in unheated samples when the protein was expressed in *E. coli* (Fig. S5c) corroborates the conclusion that the native form of the protein is a dimer.

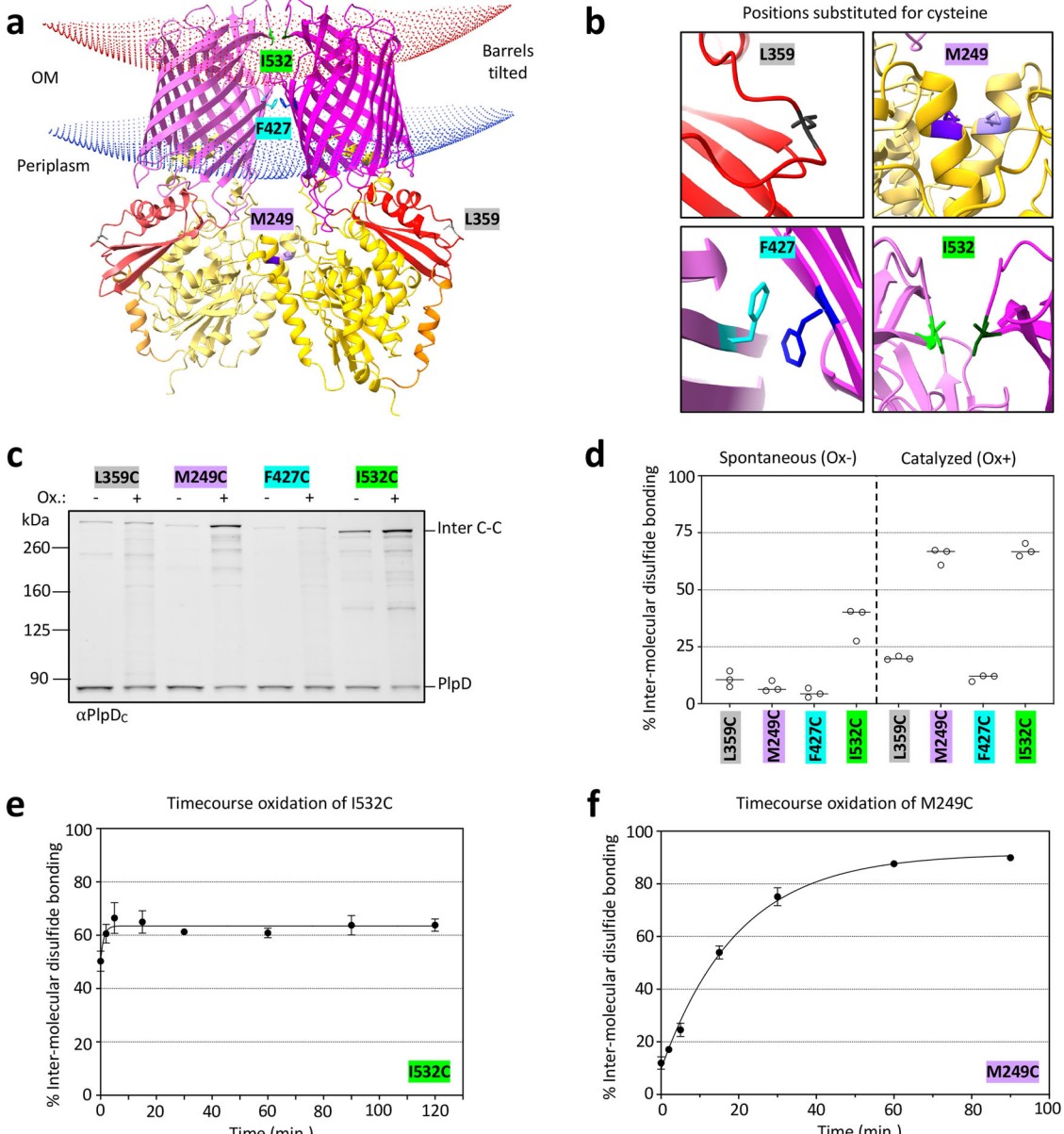

**Fig. 4 | PlpD is a homodimer. a** AlphaFold Multimer model of the PlpD dimer showing dimerization interfaces at the PL-domain and the β-barrel. Domains are color-coded as in Fig. 1a. Curving of the OM induced by dimerization of PlpD was calculated using the OPM server[50]. Residues selected to mutate to cysteine are highlighted: M249 (purple) to confirm dimerization of the PL-domains, I532 (green) and F427 (blue) to test interactions between the β-barrels, and L359 (gray) to serve as a negative control based on its location far from any predicted dimerization interface. **b** Magnified views of each selected residue illustrating its predicted proximity to the corresponding residue in the neighboring PlpD molecule. **c** SEH88 transformed with pSEH170, pSEH197, pSEH210, or pSEH213 were grown and the expression of the indicated *plpD* cysteine mutants was induced as described in Fig. 1b. Aliquots were then either mock-treated or treated with 4-DPS and analyzed by immunoblot using the anti-PlpD$_C$ antiserum. Both PlpD (~82 kDa) and an inter-molecularly crosslinked form (~300 kDa) were detected. **d** Quantitation of inter-molecular disulfide bond formation between selected cysteine residues. The levels

of disulfide bond formation in mock-treated cells ('spontaneous', Ox-) and 4-DPS-treated cells ('catalyzed', Ox +) are shown. Bars = median, *N* = 3 biologically independent samples obtained as in Fig. 3c. One-sided ANOVA and multiple comparison tests are shown in Table S5. **e** SEH88 transformed with pSEH213 were grown and the expression of the *plpD* I532C mutant was induced as described in Fig. 1b. 4-DPS was added and aliquots were removed from 0-120 mins to track the kinetics of inter-molecular disulfide bond formation. The samples were then analyzed by immunoblot using the anti-PlpD$_C$ antiserum. Curve of best fit is shown. Error bars=standard error from the mean, *N* = 3 biologically independent samples obtained as in Fig. 3c. **f** The experiment described in (**e**) was repeated except cells were transformed with pSEH197 to express the *plpD* M249C mutant and aliquots were removed from 0-90 min. Error bars and *N* are as in (**e**). Curve of best fit is shown. Representative experiments for (**e**) and (**f**) are shown in Fig. S8 and statistical analyzes are shown in Table S7.

## The PL-domain "lid" interacts dynamically with both β-barrels

Interestingly, AlphaFold predicts that a small loop that was thought to form a "lid" (residues 89-126) over the hydrophobic lipid-binding groove of the PL-domain, but that was not resolved in the crystal structure[42], instead interacts with the periplasmic side of the β-barrel lumen (Fig. 5a). To test this prediction we again used in vivo disulfide

crosslinking in SEH88. When single mutations were introduced into strand 1 of the β-barrel (F410C, R412C, L413C, L415C) or the C-terminal turn (L703C) of the β-barrel, negligible crosslinking was observed (Fig. 5b, lanes 3-4, 7-8, 11-12, 15-16, and 27-28). Because the cognate residues in the adjacent β-barrel are far away in the folded dimer, the faint bands that we observed presumably result from fortuitous

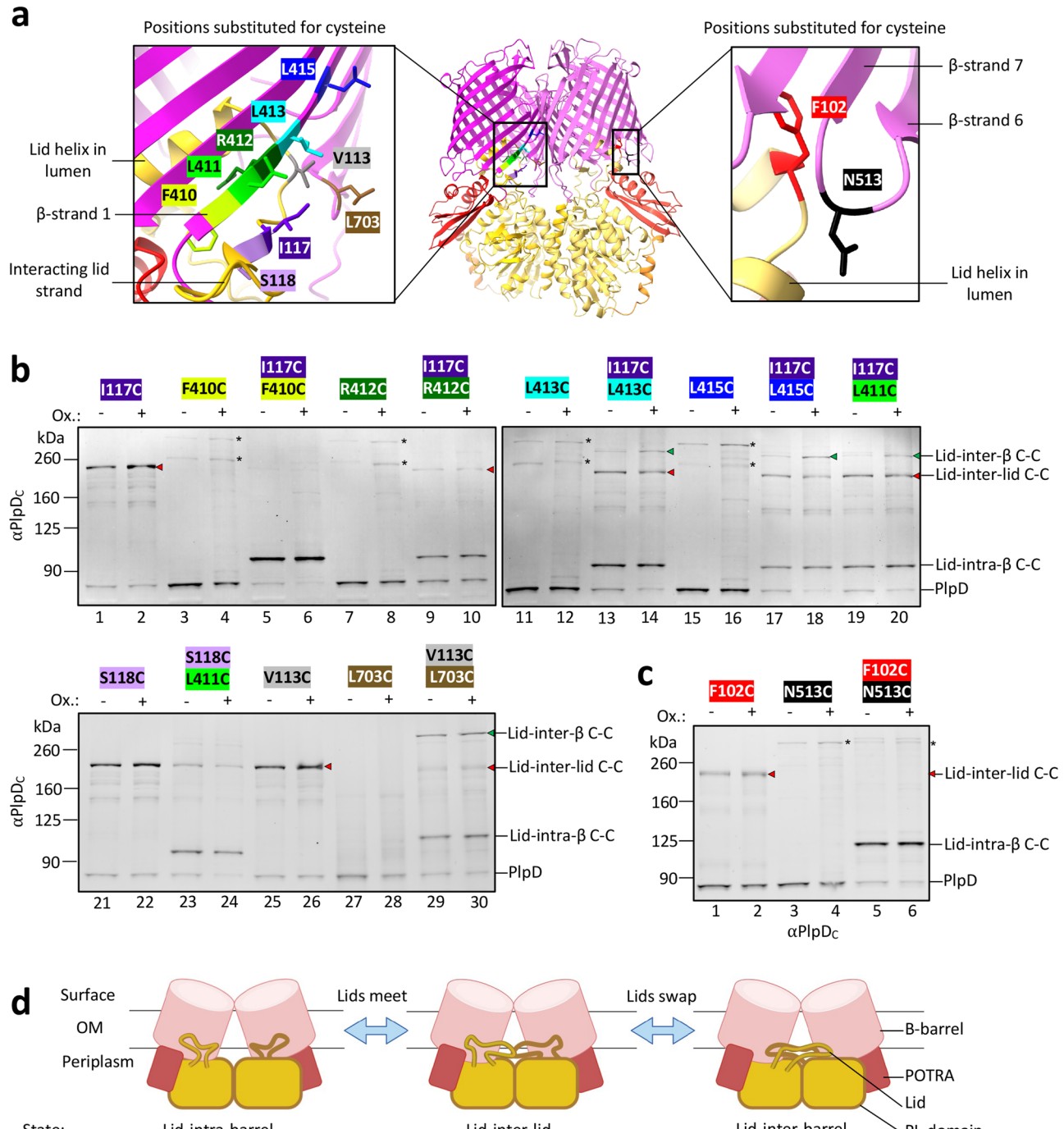

**Fig. 5 | The PlpD PL-domain "lid" is highly dynamic. a** AlphaFold Multimer model showing the PlpD dimer (middle), emphasizing the region where the PL-domain lid loops into the β-barrel and interacts with the first β-strand (magnified view, left) or the back of the β-barrel (magnified view, right). Residues selected to mutate to cysteine to probe conformational states in vivo by crosslinking are highlighted: on β-strand 1, F410 (yellow-green), L411 (green), R412 (dark green), L413 (light blue), L415 (dark blue); between β-strand 6/7, N513 (black); on β-strand 15: L703 (brown); on PL-domain, F102 (red), V113 (gray), I117 (dark purple), S118 (light purple). **b** SEH88 transformed with the appropriate plasmid were grown and the expression of the indicated *plpD* β-strand 1/15/16 and/or PL-domain lid cysteine mutants was induced as described in Fig. 1b. Aliquots were then either mock-treated or treated with the oxidant 4-DPS and samples were analyzed by immunoblot using the anti-PlpD$_C$ antiserum. The native uncrosslinked PlpD protein and intra-molecularly crosslinked and inter-molecularly crosslinked forms of the protein were detected.

The colors of the residue labels correspond to those in (a). ANOVA and multiple comparison tests between mutants treated with 4-DPS are shown in Table S8. (*) indicates non-specific crosslinking. **c** As in (b) except cysteine mutants within the lid and the far side of the β-barrel (opposite to β-strands 1/16) were investigated. (*) indicates non-specific crosslinking. **d** Cartoon representing the variable conformations of the PL-domain lid that produced the three main bands observed on the immunoblots in (b). Intra-molecular crosslinks were formed between the lid of the PL-domain and the β-barrel of the same subunit, while inter-molecular crosslinks were formed between the lids of two neighboring subunits or the lid of one subunit and the β-barrel of its neighboring subunit in the PlpD dimer. The data highlight the dynamicity of the region and the position of the lids on the periplasmic side of the β-barrel. The experiments in (b-c) were performed three times with similar results.

interactions between the two subunits in the periplasm. Surprisingly, when lid segment residues V113, I117 or S118 were converted to cysteine, a very high level of intermolecular disulfide bond formation was observed ( ~ 84%) (Fig. 5b, lanes 1-2, 21-22, and 25-26) showing that a PL-domain lid can interact with its cognate lid segment ("lid-inter-lid"). In addition, when the lid mutations were combined with a mutation in β-strand 1 of the β-barrel (F410C, L411C, R412C, L413C or L415C) or the C-terminal turn of the β-barrel (L703C), the gel migration patterns not only indicated intra-molecular crosslinks ("lid-intra-barrel") (Fig. 5b, lanes 5-6, 9-10, 13-14, 17-20, 23-24, and 29-30), but also inter-molecular crosslinks between the lid of one PlpD molecule and the barrel of the second PlpD molecule ("lid-inter-barrel") (Fig. 5b, lanes 14, 18, 20, and 29-30). The data not only confirms the predicted intramolecular interaction between the PL-domain lid with the β-barrel but also indicates that the lid slides dynamically across β-strand 1 of both β-barrels. Furthermore, we observed that lid-intra-barrel and lid-inter-lid conformations are equally abundant between some cysteine pairs even without the addition of 4-DPS (Fig. 5b, lanes 13, 17, 19), and appear to be more stable than lid-inter-barrel interactions which are significantly increased upon the addition of the oxidizing reagent (Fig. 5b, lanes 14, 18, 20). Similarly, we found that lid-intra-barrel interactions were particularly strong between I117C and F410C (lanes 5-6) and S118C and L411C (lanes 23-24). As a control to further test the unexpected lid-inter-barrel conformations, we substituted cysteines singly or in combination at positions F102 (lid) and N513 (barrel) that we predicted to be close enough to form intramolecular disulfide bonds but too distant to form lid-inter-barrel disulfides (Fig. 5c). Consistent with our hypothesis, the majority of PlpD molecules containing F102C and N513C substitutions in combination formed lid-intra-barrel disulfides.

The data support a model in which the PL-domain lid segment is highly dynamic. In addition to adopting a conformation in which the lid is intramolecularly bound to its β-barrel domain (Fig. 5d, left), it appears that the lid can unexpectedly reach across to interact with both the cognate lid (Fig. 5d, middle) as well as β-barrel β-strand 1 (Fig. 5d, right) of the neighboring PlpD subunit. The results strongly suggest that rather than being locked into a set conformation after its assembly into the OM, each PlpD homodimer frequently undergoes transitions from a lid-intra-barrel conformation to a lid-inter-barrel conformation. Interactions between cognate lid residues may, therefore, represent an intermediate state between these two major conformations. Given the proximity of the lid to the PL-domain lipid binding pocket and the association of the lid with both β-barrel strand 1 and cognate residues in the neighboring subunit, the unprecedented dynamicity of a small segment within the dimer that we observed may play an important role in the function of the PL-Omp85 family (Fig. S9).

### The membrane lipid composition is altered in a *plpD-* strain

Given that the PL-domain of PlpD and its homologs have phospholipase activity in vitro[42,43], we surmised that if the PL-domain is located in the periplasm, then it might play a role in maintaining OM lipid homeostasis. To test this possibility, we performed a comparative lipidomic analysis of the wild-type PA14 strain and the isogenic *ΔplpD* strain SEH88. Samples were collected from multiple independent cultures grown in parallel when the cells reached mid-log phase and washed with PBS before further processing. Remarkably, the two strains showed a complete separation in a principal component analysis (Fig. 6a). In addition, a volcano plot and a heat map showed that the level of a wide variety of lipids was considerably higher in the deletion strain than in the wild-type strain (Fig. 6b, c; Fig. S14). This group included glycerophospholipids [phosphatidic acid (PA), phosphatidylglycerol (PG), phosphatidylserine (PS), phosphatidylethanolamine (PE) and cardiolipin (CL)] that are the major components of bacterial membranes, lysophospholipids that are thought to play roles in membrane remodeling[55] and other lipids. Although the exact

function of PlpD cannot be determined from these results, the data support the idea that the protein acts as a phospholipase that cleaves endogenous lipids and that, in its absence, a subset of lipids accumulate. Furthermore, the data strongly corroborate other evidence that the PL-domain is located in the periplasm because it is difficult to imagine how an OMP would influence the level of membrane lipids that are found predominately in the inner leaflet of the OM if its active site were located on the cell surface.

## Discussion

In this study we provide insights into the structure of the *P. aeruginosa* OMP PlpD that challenge long-standing assumptions about the orientation and oligomeric state of the PL-Omp85 family. In contrast to a previous report that classified PlpD as the archetype of a type Vd secretion system in which the PL-domain is secreted[41], we did not find the PL-domain in the culture medium when we expressed the protein in either a highly pathogenic strain of *P. aeruginosa* or in *E. coli*. This observation led to a series of experiments in which we used PlpD truncations and evidence that the *P. aeruginosa* OM is highly permeable to exogenous proteases to demonstrate that the PL-domain is located in the periplasm. Interestingly, AlphaFold predicts with high confidence that the PL-domain of PlpD and its homologs resides in close association with the POTRA domain, a segment of Omp85 proteins that has only been found in the periplasm. This prediction, which was supported by in vivo intramolecular disulfide bond formation assays, provided further evidence that the PL-domain remains on the periplasmic side of the OM. Consistent with this conclusion, a *Fusobacterium nucleatum* homolog of PlpD (FplA) recently was not detected on the cell surface[44]. We also obtained strong evidence that unlike other characterized members of the Omp85 superfamily, PlpD is a dimer in which a dynamic lid that extends from the PL-domain catalytic pocket moves between the two β-barrels of the dimer. To the best of our knowledge, this type of dynamicity has not been previously observed in a membrane protein.

Our data strongly suggest that the homodimeric form of PlpD in which the PL-domain is located in the periplasm represents the final structure of the protein. An alternative model in which the folded PL-domain is translocated across the OM when cells receive an unidentified biological cue appears to be energetically and structurally highly unfavorable. In order to be translocated to the extracellular space the folded and dimerized PL-domains would first need to dissociate, thereby exposing the hydrophobic intermolecular interface to the periplasm. The 31 kDa PL-domains would then presumably need to be unfolded (in an environment that lacks ATP) to be secreted through a β-barrel that has a relatively small pore size. In this regard, it is notable that the β-barrels of "autotransporters" (type Va and Vc pathways) play a role in the secretion of a covalently linked extracellular passenger domain at a relatively early stage of assembly when they form an open hybrid barrel with BamA[51,56], but have never been observed to promote passenger domain secretion once they are fully folded. Single passenger domains are secreted as a largely unfolded polypeptide that folds slowly in the extracellular space due to its unusual β-helical structure, whereas the three passenger domains of trimeric autotransporters fold coordinately into a coiled-coil structure[57]. The so-called exoproteins of two-partner secretion (type Vb) pathways are secreted through a fully folded β-barrel protein that is a member of the Omp85 superfamily, but like the passenger domains of classical autotransporters are secreted by undergoing a transition from an unfolded state to a β-helix in the extracellular milieu[39].

Although none of the previously studied members of the Omp85 superfamily found in bacteria are oligomers, a mitochondrial homolog of BamA called Sam50 was recently shown to form a potentially transient homodimer[57]. Like BamA, however, Sam50 catalyzes the assembly of OMPs, and it has been hypothesized that in the

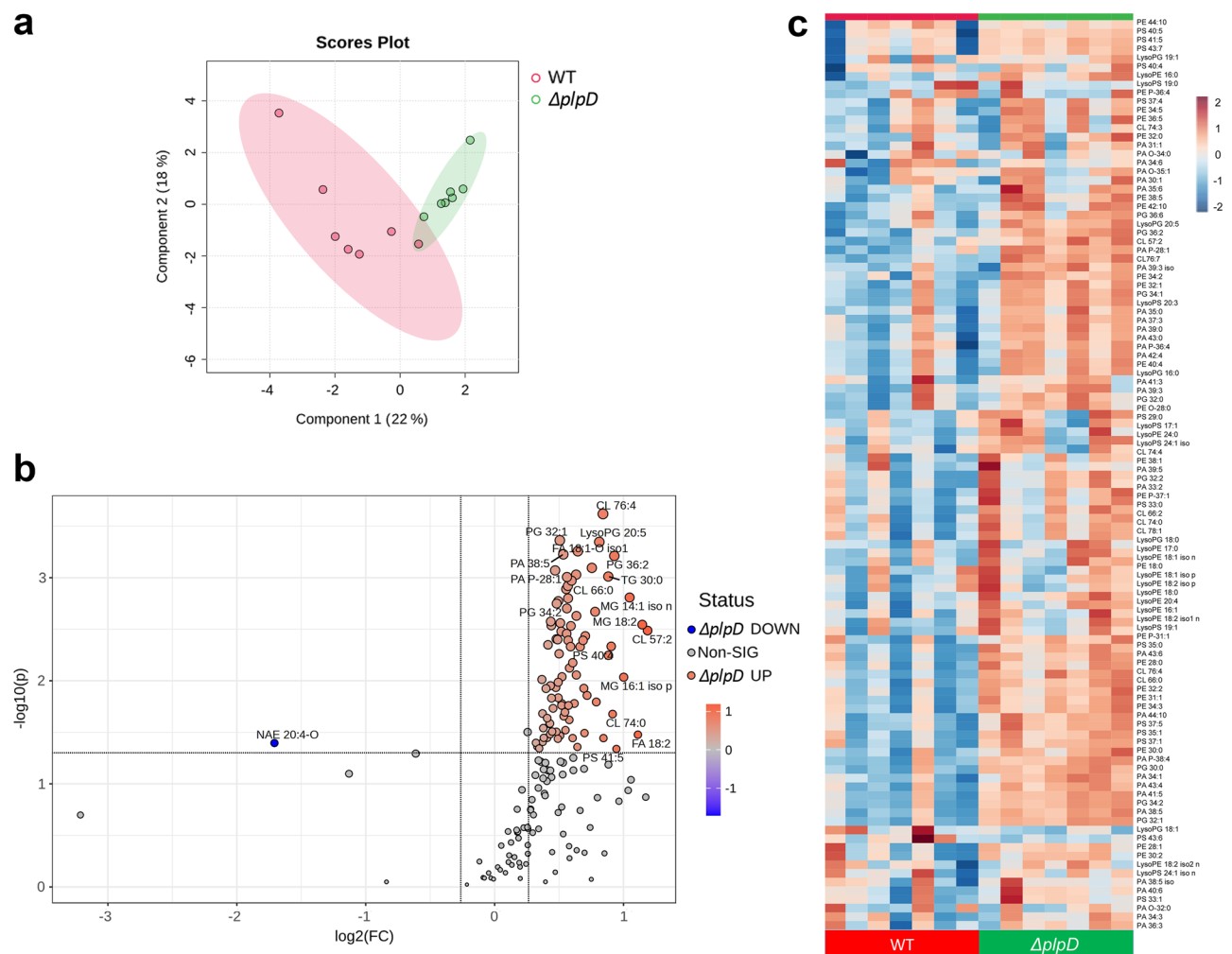

**Fig. 6 | The level of phospholipids is globally elevated in a *ΔplpD* strain.** PA14 (WT) and SEH88 (*ΔplpD*) cultures were grown in parallel in LB and washed in PBS before the cells were lysed and prepared for a lipidomic analysis using LC/MS. **a** A principal component analysis (PCA) of the lipid profiles is shown. PA14 profiles are shown in red and SEH88 profiles are shown in green. **b** An unpaired, parametric Welch *t*-test (two-sided) with unequal variance was used to generate a volcano plot. Lipids that were present at a higher level in SEH88 are shown in red and lipids that were present at a lower level in SEH88 are shown in blue. Log 2(FC) refers to log2 for fold change (FC) with a cut-off line for FC ≥ 1.2; -log(p) refers to -log10 for p-values with a cut-off line for p-value ≤ 0.05. **c** A heatmap showing the phospholipids that were differentially produced in PA14 and SEH88 cultures was generated using Pearson-Ward correlation analysis. The level of the differential for each lipid is shown in Table S9, a detailed list of the LC/MS mass assignments is shown in Table S10, and a heatmap showing the top 70 lipids that were differentially produced is shown in Fig. S14. The PCA, volcano plot and heat map were generated using MetaboAnalyst 5.0[70]. Abbreviations: cardiolipin (CL); fatty acid (FA); oxidixed fatty acid (FA-O); monoacylated glycerophospholipid (Lyso); monoglycerol (MG); phosphatidic acid (PA); phosphatidylethanolamine (PE); phosphatidylglycerol (PG); phosphatidylserine (PS).

yeast *S. cerevisiae*, the second barrel may be responsible for releasing an incoming protein from the Sam50 complex and/or serving as a place holder in the complex until it is replaced by a newly folding OMP[58]. Perhaps more significantly, the β-barrels of the Sam50 dimer are in a partially open confirmation that presumably is required for their function in OMP assembly[58]. The chloroplast Omp85 protein Toc75 and another subunit of the TOC complex have also recently been shown to form a hybrid barrel that has been proposed to facilitate protein translocation[37,38]. In contrast, PlpD and its homologs appear to lack a lateral gate that might widen the dimer to allow PL-domain translocation. Curiously, a small portion of the first strand of the PlpD β-barrel binds to a lid sequence (Ile-Ser-Phe) that is similar to the last three residues of the "β-signal", a motif that is found at the C-terminus of the majority of OMPs and bound by the first strand of the BamA β-barrel during OMP assembly[22,24,59]. The lid-associated β-signal motif is conserved across multiple PL-Omp85 family members (Fig. S13), which suggests that β-signals have a broader function in binding to Omp85 β-barrel domains.

Although the exact physiological function of PlpD and its homologs is unknown, it is striking that we found that the relative levels of many different phospholipids were substantially higher in a *ΔplpD* strain than in a wild-type strain. Our results are consistent with the observation that the PL-domains of PlpD and FplA (expressed without the β-barrel and POTRA domains) bind to a variety of phospholipids and have a phospholipase $A_1$ activity that, like that of other phosphatases, is dependent on a Ser-Asp catalytic dyad[43]. The lipid differential that we discovered strongly suggests that the PL-domain of PlpD cleaves multiple phospholipids, although we cannot determine if the elevation in the levels of some of the lipids in the *ΔplpD* strain is a secondary effect of cellular response to the loss of PlpD. For example, under normal conditions bacterial cardiolipin synthase uses PG as a substrate to produce CL[60], so an increase in PG levels in a *ΔplpD* strain might lead to an increase in CL. Because phospholipids are largely excluded from the outer leaflet of the *Pseudomonas* OM[61] it is very likely that the PlpD PL-domain would have to reside in the periplasm to cleave endogenous

phospholipids. Furthermore, the notion that the PL-domain functions intracellularly is supported by evidence that neither PlpD nor FplA appears to function as a secreted virulence factor that cleaves lipids in the host. A study on the effects of purified PlpD and FlpA injected into *G. mellonella* larvae reported no direct toxic effects, even at very high concentrations[43]. In addition, patatin-like phospholipases produced by bacteria (e.g., *Vibrio cholera*) have previously been shown to function in an intracellular location[62].

Indeed the dimerization of PlpD and its homologs, the intracellular localization of the PL-domain, and our lipidomic analysis strongly suggest that PL-Omp85 proteins function in OM lipid homeostasis, possibly by binding to and processing phospholipids in the inner leaflet of the OM. Outer membrane phospholipase A (OMPLA or PldA), a phospholipase involved in OM lipid homeostasis in *E. coli*, is the only other β-barrel protein known to dimerize in the bacterial OM[63]. Dimerization is necessary to trigger phospholipase activity in OMPLA by bending the OM to sequester a pair of target outer leaflet lipids between the two subunits of the dimer[64,65]. PlpD dimerization also induces mutual tilting of the β-barrel domains, but in the opposite direction from OMPLA (Fig. S7). The negative curvature created in the inner leaflet of the membrane space between the PlpD β-barrels, coupled with the presence of the amphipathic lid region that contains an aromatic residue that fluctuates between the two β-barrels in this space, strongly suggests a model in which the function of dimerization is to create an unstable inner leaflet zone. Structural and molecular dynamics studies strongly support a function of the Omp85 proteins BamA and TamA in membrane destabilization near the first and last β-strands of the β-barrel domain which has been proposed to create an energetically favorable membrane environment for OMP integration[20,21,23,66]. In the case of PL-Omp85 proteins, however, the predicted positioning of the PL-domain directly under the zone of membrane destabilization might instead create an energetically favorable environment for lipid extraction directly into the PL-domain lipid binding pocket (Fig. S10). Regardless of the function of PlpD and its homologs, we speculate that the unprecedented dynamicity of the PL-domain lid region, which is able to reach across the gap between the two subunits and interact with the neighboring lid or β-barrel domain, plays an important role in regulating their activity. One can certainly imagine a scenario in which membrane changes (perhaps sensed by the β-barrel domains) promotes or inhibits movement of the lids and thereby turns the enzymatic activity of the PL-domain on or off.

## Methods

### Bacteria and growth conditions
*E. coli* B strain BL21(DE3) and SEH88, a derivative of *P. aeruginosa* strain PA14 (see below) were used in all experiments. *E. coli* transformed with appropriate plasmids were grown to $OD_{600}$ ~ 1 in lysogeny broth (LB, Miller) containing 50 µg mL⁻¹ trimethoprim at 37 °C/shaking at 250 rpm and the production of PlpD (or derivatives) was induced for 30 min by the addition of 0.2% L-rhamnose. *P. aeruginosa* cells were grown in LB containing 100 µg mL⁻¹ trimethoprim as described above to $OD_{600}$ ~ 0.2–0.3 and the production of PlpD (or derivatives) was induced with 0.2% L-rhamnose for 3 h.

### Plasmid and strain construction
The plasmids used in this study are listed in Supplementary Table 1, and the oligonucleotides and dsDNA gene blocks used to construct plasmids are listed in Supplementary Tables 2 and 3, respectively. To construct pMTD1523, pMTD607[22] was amplified by PCR with the primers mtd350 and mtd351 and subsequently assembled by Gibson Assembly with the dsDNA fragment mtd359 encoding an *E. coli* codon optimized *plpD* gene (PL-domain deleted). To construct additional PlpD truncations for expression in *E. coli*, pMTD1523 was PCR amplified using primers seh4/5 (to remove the linker) or seh4/7 (to remove the

POTRA domain) and then re-circularized by ligation. To re-introduce the PL-domain to pMTD1523, the plasmid was PCR amplified using seh13/14 and assembled with dsDNA fragment mtd372. To generate a plasmid for the expression of the full-length native PlpD protein with an associated N-terminal StrepII tag in *P. aeruginosa* PA14 (pSEH81), the dsDNA gene blocks seh15/16 were combined via Gibson Assembly with NdeI-digested pSCrhaB2[46]. Truncations and point mutations were generated by amplifying pSEH81 with primers seh35/36 (to remove the PL-domain), seh35/37 (to remove the linker), seh35/38 (to remove the POTRA domain) or appropriate mutagenesis primers. To construct pSEH306, pSEH81 was amplified using the primers seh206 and seh207 (to remove the TS tag) and then re-circularized. To delete the chromosomal copy of *plpD* in PA14, the suicide plasmid pDONRPEX18Gm[67] was linearized by PCR amplification with seh17/18 and assembled with dsDNA fragment seh23 containing chromosomal sequences flanking *plpD* to generate pSEH78. PA14 was then transformed with pSEH78 to create a scarless *plpD* gene deletion strain (SEH88) by following a previously described protocol[68].

### PlpD topology assays
To monitor the surface exposure of the PlpD phospholipid binding domain, PlpD was produced in *E. coli* or *P. aeruginosa* as described above. Cells (1 mL *E. coli* or 0.5 mL *P. aeruginosa*) were pelleted (10,000 x *g*, 2 min, 4 °C), resuspended in ice cold PBS, and incubated on ice for 20 min (*E. coli*) or 30 min (*P. aeruginosa*) with 200 µg mL⁻¹ proteinase K (PK) or with PK buffer (5 mM $CaCl_2$, 50 mM Tris-HCl pH 8) for the mock-treated control, or 200 µg mL⁻¹ chymotrypsin. Cells were then pelleted (10,000 x *g*, 2 min, 4 °C for *E. coli* or 16,000 x *g*, 2 min, 4 °C for *P. aeruginosa*), resuspended in ice cold PBS, and incubated with 4 mM PMSF and 10% (v/v) TCA on ice for 10 min to inhibit PK and precipitate proteins. Precipitates were pelleted (20,817 x *g*, 10 min, 4 °C), washed with 0.6 mL acetone, pelleted again, and air dried at 37 °C for 20 min. Dried precipitates were resuspended in 2x SDS protein gel loading buffer (Quality Biological) in a volume normalized to the final culture $OD_{600}$ reading (volume, µL = $OD_{600}$ x 200 for *E. coli* or $OD_{600}$ x 100 for *P. aeruginosa*) and heated at 99 °C for 20 min. In some experiments cells were collected as described above but permeabilized by resuspending them in spheroplast buffer (40% sucrose, 33 mM TrisHCl pH 7.4) and incubating them on ice for 20 min with 200 µg mL⁻¹ lysozyme and 2 mM EDTA prior to the addition of PK. PlpD truncations purified from *E. coli* were resuspended to 150 µg/mL and treated for 0-30 min with 10 µg/mL chymotrypsin. After the appropriate time, proteins were precipitated and processed as above (but washed with 200 µL acetone). Dried proteins were resuspended in 10 µL protein gel loading buffer before being boiled and separated by SDS-PAGE.

### Disulfide-bond formation assay
We used a modified version of a previously described protocol to analyze site-specific intra- and intermolecular protein interactions[51]. PlpD expression was induced in *P. aeruginosa* and 1 mL samples were aliquoted into 1.5 mL tubes on ice. Cells were then pelleted (10,000 x *g*, 2 min, 4 °C), resuspended in 1 mL ice-cold PBS, and incubated with the thiol-specific oxidizer 4,4'-dipyridyl disulfide (4-DPS) at a final concentration of 0.2 mM (or an equivalent volume of ethanol for mock-treated controls) for 30 min. Cells were then pelleted (16,000 x *g*, 2 min, 4 °C), resuspended in 0.5 mL ice-cold PBS, and mixed with PMSF and TCA as described above. To monitor the kinetics of intermolecular disulfide-bond formation, 5 mL samples of induced subculture were aliquoted into 50 mL tubes on ice, pelleted (3000 x *g*, 5 min, 4 °C), and washed with 10 mL ice-cold PBS. Cells were then resuspended in 5 mL ice-cold PBS and incubated with 0.2 mM 4-DPS for 0, 2, 5, 15, 30, 60, 90, and 120 min. At each time point 0.4 mL aliquots were dispensed into 1.5 mL tubes on ice pre-loaded with PMSF and TCA. Precipitates were washed and mixed with 2x SDS protein gel loading buffer as described above.

## Western immunoblotting, imaging, and quantitation

Proteins were separated by SDS-PAGE on 8-16% Tris-glycine gels (Thermo Fisher catalog number XP08162BOX) and transferred to nitrocellulose membranes using an iBlotII (Life Technologies). Blots were probed overnight using a mouse monoclonal anti-StrepII antibody (Qiagen catalog number 34850) at a 1:2500-1:5000 dilution or a rabbit polyclonal antiserum raised by Covance Research Products against the HPLC purified PlpD C-terminal peptide $NH_2$-GINDENFKA-$^{FY}$LNLGQNC-COOH at a 1:5000 dilution. Blots were subsequently probed with a fluorescent goat anti-mouse (IRDye 800CW) or goat anti-rabbit (IRDye 680LT) antisera (LICOR catalog numbers 926-32210 or 926-68021, respectively). The membranes were then scanned using an Amersham Typhoon 5 imager (GE Healthcare) with a 685 nm laser (IRshort720BP20 filter) or 785 nm laser (IRlong825BP30 filter) and the PMT set at 450 V or 700 V, respectively. Pixel intensities of detected proteins were measured using Fiji software (v2.9.0/1.53t). Within-lane values were used to calculate the percent intra-molecular disulfide-bond formation [($^{intra}$PlpD/ ($^{intra}$PlpD + $^{inter}$PlpD + $^{FL}$PlpD)) x 100] or the percent stable fragment resulting from chymotrypsin digestion. GraphPad Prism 9 was used for all statistical analyzes. The anti-PlpD C-terminal peptide antiserum was validated by showing that a protein band of the expected molecular weight could only be detected in *E. coli* transformed with pMTD1523 after the expression of *plpD* was induced. Uncropped images of all blots are included in the Source Data file.

## Heat-dependent protein mobility shift assay

Bacteria were collected as described above for protease digestions, resuspended in BugBuster Master Mix (EMD Millipore catalog number 71456) (volume, $\mu$L = $OD_{600}$ x 150) and lysed on ice for 3 min. Aliquots (15 $\mu$L) of lysates were mixed with 5 $\mu$L 2x SDS protein gel loading solution to bring the final SDS concentration to 1%. Samples were either maintained on ice or heated to 99 °C for 20 min. Proteins were then resolved by 'cold' SDS-PAGE (i.e., by packing gel tanks in ice and running the gels in a cold room) and transferred to nitrocellulose for immunoblotting.

## Protein purification

*E. coli* strain BL21(DE3) transformed with pMTD1523, pSEH58, or pSEH63 was grown overnight at 37 °C, shaking at 250 rpm in LB containing 50 $\mu$g/mL trimethoprim. The cells were pelleted (4000 x *g*, 15 min, 4 °C) and resuspended in the original volume of LB. Four aerating flasks containing 1 L culture medium were each seeded to OD = 0.05. Subcultures were incubated at 37 °C, 300 rpm, grown to OD ~ 1, and the expression of PlpD derivatives was then induced with 0.2% L-rhamnose for 3 h. Cells were then pelleted (5000 x *g*, 10 min, 4 °C), washed in 25 mL ice-cold PBS, and resuspended in 50 mL ice cold PBS containing 1 mM EDTA and EDTA-free SigmaFast protease inhibitors (Sigma Aldrich catalog number 8820). Cells were lysed using a Constant Systems BT40 cell disruptor and unbroken cells were pelleted (20,000 x *g*, 15 min, 4 °C). The supernatant was then centrifuged (266,112 x *g*, 40 min, 4 °C) to isolate cell membranes. Membrane pellets were gently washed with 10 mL ice cold PBS and then homogenized into 100 mL TN buffer (25 mM TrisHCl, 300 mM NaCl, pH 8) with a Dounce homogenizer. The membranes were pelleted as before, supernatant removed, and then membranes were homogenized into 96 mL solubilization buffer (25 mM TrisHCl, 300 mM NaCl, 1 mM EDTA, 1% DDM (w/v), pH 8) including EDTA-free SigmaFast protease inhibitors) and incubated at 4 °C for 4 h with constant rotation. The solution was centrifuged (266,112 x *g*, 40 min, 4 °C) to remove insoluble material and the supernatant was transferred to a flask containing 10 mL Streptactin beads equilibrated in TN-DDM (25 mM TrisHCl, 300 mM NaCl, 0.03% DDM, pH 8) and rotated at 4 °C overnight. The next day, the beads were collected in a gravity flow column and were washed 13x with 30 mL TN-DDM. The proteins were eluted in eight 5 mL fractions of biotin buffer (50 mM biotin, 25 mM TrisHCl, 300 mM

NaCl, 0.03% DDM). All of the fractions were pooled and subsequently concentrated to a volume of 2.5 mL in a Pierce Protein Concentrator PES, 10 K MWCO. The concentrated protein was exchanged into TN-DDM using a PD-10 desalting column. PlpD derivatives were then concentrated again until the volume measured less than 0.5 mL. The concentrated protein was aliquoted, frozen in liquid nitrogen, and stored at −80 °C.

## Biophysical assays

Circular dichroism spectra of purified PlpD fragments were determined using a JASCO J-715 spectropolarimeter. Concentrated protein was diluted to 0.1 mg/mL in NaPB (10 mM sodium phosphate, pH 7, 0.03% DDM), and loaded into 0.5 mm quartz cuvettes immediately after a baseline spectrum was obtained from NaPB (range: 190-260 nm, bandwidth: 1.0 nm, integration time: 1 s, scanning speed: 20 nm/min). For tryptophan fluorescence, 0.75 $\mu$M purified PlpD fragments in 4 M G-PBS-DDM (4 M guanidine HCl, 1x PBS, 0.03% DDM) were incubated at 20 °C for 23 h and aliquoted in triplicate into a clear-bottomed, black 96-well plate (200 $\mu$L/well) alongside freshly prepared solutions of 0.75 $\mu$M protein in PBS-DDM (1x PBS, 0.03% DDM). Plates were equilibrated at 20 °C for 1 hr and loaded onto a ThermoFisher Varioskan LUX multimode microplate reader using an excitation wavelength of 295 nm (5 nm bandwidth) and an emission range of 313-400 nm.

## Lipidomic analysis of *P. aeruginosa* strains

Ten PA14 and SEH88 overnight cultures were grown in LB, pelleted (3000 x *g*, 10 min, 4 °C), resuspended in 10 mL fresh LB, and each diluted into a 50 ml culture at $OD_{600}$ = 0.05 in 125 ml flasks. The cultures were grown for 3 h at 37 °C in a shaking water bath (at 250 rpm). After determining the final $OD_{600}$, 33.13 $OD_{600}$ equivalents were removed from each culture and the cells were pelleted (4000 x *g*, 10 min, 4 °C). The cells were washed twice by resuspending them in 30 ml ice cold PBS and pelleting them (4000 x *g*, 10 min, 4 °C), and then resuspended in 1 ml ice cold PBS. Cells were pelleted again (16000 x *g*, 2 min, 4 °C) and the supernatants were removed. Cell pellets were then frozen on dry ice and stored at −80 °C until further use.

For lipidomic analyzes, each cell pellet was resuspended in 200 mL chilled MilliQ water. Cells were lysed using a Misonix XL-2000 Ultra-liquid processor sonicator at 40 amps for 0.5 min and incubated on ice. Cell lysates were vortexed and then a 20 mL aliquot was snap-frozen on dry ice and stored at −80 °C to conduct Bradford protein quantification for later sample normalization[69-71]. An adapted Bligh and Dyer biphasic liquid extraction[70,71] was conducted using a 2:2:1 chloroform ($CHCl_3$)/methanol (MeOH)/water ratio. $CHCl_3$ (100 $\mu$L) spiked with qIS, 0.06 $\mu$g/mL phenyl-N-pyridinyl acrylamide (PNPA) was added to each extract. Samples were vortexed on a BenchMixer (Benchmark Scientific) at speed 6 for 15 s and then incubated on ice for 20 min. The samples were centrifuged (12,000 x *g*, 15 min, 4 °C) to generate two phases (an upper hydrophilic and a lower hydrophobic lipid phase) and to discard the remaining protein layer. The hydrophobic phases were concentrated under a $N_2$ gas vapor stream until they were completely dry (~2 h), snap-frozen on dry ice and stored at −80 °C until the time of analysis.

Only LC-MS grade solvents and additives (Covachem) were used to prepare reagents, mobile phases, and wash solutions unless otherwise indicated. All hydrophobic extracts were reconstituted in 100 $\mu$L 5:4:1 ethanol (EtOH)/MeOH/water and randomized during extraction as well as prior to analysis using an Agilent 6545 Quadrupole time-of-flight mass spectrometer with Agilent Infinity II 1290 Ultra High-Performance Liquid Chromatography. The lipids along with other metabolites were resolved on a reverse-phase Acquity CSH 2.5 $\mu$m, 2.1 × 100 mm column (Waters) utilizing a gradient composed of mobile phase A, 70:30 Water/ Acetonitrile (MeCN) with 5 mM ammonium formate (aq) + 0.1% formic acid (FA) and mobile phase B, 90:10 iso-propanol/ MeCN with 5 mM ammonium formate (aq) + 0.1% FA. An

isothermal column temperature of 40 °C and a static flow rate of 0.200 mL/min was maintained. Real-time mass correction was applied with a 0.18 mL/min infusion of an external standard (containing TFA/PURINE/HP921) in 95:5 MeCN/water. Electrospray injection (ESI) negative (-) ion acquisition was applied with the following MS parameters: injection volume, 8 μL; drying gas temperature (temp), 175 °C; drying gas flow, 8 L/min; nebulizer pressure, 45 psi; sheath gas temp, 350 °C; sheath gas flow, 12 L/min; capillary voltage, 3000 V; nozzle voltage, 25 V; fragmentor, 90 V; skimmer, 50 V; scan rate, 3.0 spectra/s; mass range m/z 100–2850. The following gradient timetable was applied: 0–0.1 min, 1% B; 0.8 min, 5% B, 2 min 35% B; 4 min, 38%; 4.25 min, 40% B; 7 min, 98% B; 7.5 min, 99%; hold, 0.75 min; 8.75 min; 35%; 10.5 min, 90% B; 11.5 min, 38% B; 12.5 min, 1%; equilibrate, 1.5 min. Alternatively, ESI positive ion acquisition applied the following MS parameters: injection volume, 6.5 μL drying gas temperature (temp), 250 °C; gas flow 8 L/min; nebulizer, 45 psig; sheath gas temp, 350 °C; sheath gas flow, 12 L/min; capillary voltage, 3500 V; nozzle voltage, 1000 V; fragmentor, 170 V; skimmer, 50 V; scan rate, 3.5 spectra/s; mass range, m/z 100–2850. During ESI +, following gradient timetable was applied: 0–0.1 min, 1% B; 0.8 min, 5% B, 2 min 35% B; 4.5 min, 38%; 4.75 min, 40% B; 7.5 min, 71% B; 8.5 min, 80.5% B; 9.5 min, 81.5%; 11 min, 98% B; 11.25 min, 99% B; 11.25 min, 99%; 12.25 min, 100% B; hold 0.25 min; 13 min, 75% B; 14 min, 90% B; 15 min, 1% B; equilibrate, 1.5 min.

Prior to pre-processing each dataset, the extracted ion and total ion chromatograms (XIC and TIC, respectively) were examined for the pooled quality control (QC) samples to inspect the consistency of retention time and ionization levels. Following data acquisition, the *P. aeruginosa* Metabolome Database (PAMDB) version 1.0 (http://pseudomonas.umaryland.edu), and the *E. coli* Database (Ecocyc.org) including a mass database for the taxonomy of known *P. aeruginosa* strains were used as references to assign putative identifications for the mass features detected in the pooled QC sample along with follow-on Personal Compound Data Library (PCDL) [66] development. Lipid Maps (*lipidmaps.org*) was also used to expand the library of lipid species including diacylglycerols (DG), monoacylglycerols (MG), triacylglycerols (TG), cardiolipin (CL), phosphatidylethanolamine (PE), phosphatidylserine (PS) and phosphatidylglycerol (PG) along with its phosphatic acid (PA) precursor that are found in the membranes of *P. aeruginosa* and other bacilliform bacteria. The two resulting ESI+ and ESI- CSH lipidomic gradient-specific PCDL were utilized to perform targeted mass ion selection and alignment parameters restricted for logical binning of the input data to ion mass range ± 2.0 mDa and retention time ± 0.4 min. The mass selections were limited to features that were either protonated ( + H) for ESI+ or deprotonated(-H) for ESI- without including other adducts or neutral losses. These parameters were used to select and annotate detected mass features and to define mass feature bins which allowed partitioning of the m/z versus retention time (RT) matrices into fixed widths utilizing targeted PCDL-based selection via Agilent Masshunter Profinder B.08.00. The bins were inspected manually to evaluate consistency of integration for each compound across all samples. Following pre-processing, the mass ion peak area for each compound was corrected to the area of sample-dependent qIS. Bradford protein quantification was used to determine the protein concentration in each sample and normalize the peak areas across all detected masses[67–69].

For multivariate analysis, data from ESI- and ESI+ analyzes were merged, log-transformed and scaled to the mean center before generating score plots, heatmaps and a volcano plot with Pearson-Ward correlation analysis[72]. In addition, the unpaired, parametric Welch *t*-test with unequal variance was performed to generate volcano plots and the biostatistical analysis within the MetaboAnalyst version 5.0 platform[72].

## Reporting summary

Further information on research design is available in the Nature Portfolio Reporting Summary linked to this article.

## Data availability

The data that support this study are available from the corresponding authors upon request. All data generated or analyzed in this study are contained within the published article, the Supplementary Information, or the Source Data files, or have been deposited in the Metabolight comprehensive data repository [https://www.ebi.ac.uk/metabolights/] under accession number MTBLS9803. Raw data for the lipidomics analysis can also be obtained from Mioara Larion (mioara.larion@nih.gov) upon request. The sequences of the proteins described in this study can be accessed through the UniParc component of the UniProt database (https://www.uniprot.org). Source data are provided with this paper.

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

## Acknowledgements

We would like to thank Xu Wang for critical reading of the manuscript. This work was supported by the Intramural Research Programs of the National Institute of Diabetes and Digestive and Kidney Diseases and the National Cancer Institute.

## Author contributions

The study was conceived by M.T.D. The experiments were designed by S.E.H., M.T.D., and H.D.B. Except for the lipidomics analysis (which was performed by T.D. and M.L.), all other experiments were conducted by S.E.H. and M.T.D. The manuscript was written by S.E.H., M.T.D. and H.D.B.

## Funding

## Competing interests

The authors declare no competing interests.
