## [Peer Review File · Nature Communications]

The patatin-like protein PlpD forms structurally dynamic homodimers in the *Pseudomonas aeruginosa* outer membraneReviewer #1 (Remarks to the Author):

In this well-written manuscript, Hanson and colleagues report the investigation of *P. aeruginosa* OMP PlpD and provide new insights about its membrane orientation and quaternary structure. The results in this manuscript challenge the standing view that type Vd proteins: (1) are OM proteins that display /secrete the functional domain outside of the bacterial cell (similar to other Type V proteins), and (2) are monomeric proteins (similar to other OM beta barrel proteins).

Based on bacterial expression of PlpD wt and mutants, the latter designed informed by an alpha fold model of PlpD, combined with cross-linking experiments analyzed by immunoblotting, the authors propose that PlpD is homodimeric with the PL-domains located in the periplasm.

This is really exciting and although the experiments are logical, given the implications of the proposed paradigm change for PlpD specifically and Type Vd proteins in general, this work would benefit from additional supporting data to support the claims.

Specific comments include the following:

1. The long N-terminal signal sequence is key for the transportation and folding of all Type5 proteins. I worry about the addition of the TwinStrepII (TS) tag (28 aminoacids) at the N-terminus of the PL-domain. This could alter the processing and translocation of the protein across the OM. Was the experiment conducted with the un-tagged protein and detected with the antiserum generated against a C-terminal PlpD peptide (aPlpDC) to see whether full length or processed protein (without the PL domain) was present in the membrane fraction?

2. It would be important to show that the purified recombinant proteins carrying the TS tag, are fully folded. Some data is presented in supplementary figures 4, which is not discussed or mentioned in the manuscript.

Figure 4S data is crucial to support the rest of the manuscript and the authors could add some missing information

(i) The Coomassie stained SDS-PAGE gel of the purified proteins corresponding to the samples used for biophysical characterisation should be included. To confirm that the beta-barrel is folded in these constructs I would suggest including a gel with samples both, boiled and not boiled. The Heat-dependent protein mobility shift assay is mentioned in the methods section but no related data is shown in the manuscript.

(ii) Ideally the recombinant production and purification of the full length PlpD should be presented, given that this is the protein that is proposed to have a different 3D structure and oligomeric state to other type V proteins. The secondary structure checked by CD, apparent MW determined by biophysical techniques such as DLS, analytical ultracentrifugation, size exclusion chromatography (perhaps using one of the deletion mutants (i.e. monomeric LPB) as control). If not enough full length recombinant protein can be obtained for purification, the authors can consider analyzing the native and all relevant mutants using heat-dependent protein mobility shift assay and native PAGE gels.

3. Fig 1b, The authors indicate that they detect a band corresponding to full-length PlpD (~82 kDa) in cell pellet samples. Could this indicate that the protein is not correctly folded and therefore not translocated and processed.

Fig 1c. The authors state that in *Pseudomonas* the protease is able to permeabilise the OM. in this case the +/- permeabilization experiment seems redundant since it cannot inform about whether the

PL domain has been translocated. These experiments show that the full-length protein including the TS tag is present in the pellet which could be because the TS tag impedes correct translocation outside of the cell. I suggest carrying out the experiment with an untagged construct or not tagged at the n-terminus.

4. Figure 3. It is unclear why the formation of an intramolecular disulphide bond upon mutation of the predicted interacting residues into C and thiol oxidation with 4-DPS would lead to increasing the MW on SDS-PAGE.

What is the apparent MW of the new band? Were these samples also run in the presence of a reducing agent and did the higher MW band disappear. I would also suggest testing the un-boiled proteins in parallel. Were the single mutants in the Potra domain Q383C and V388C tested? Could the experiment be trapping a complex between PlpD and another periplasmic protein?

5. The dimerization of PlpD is very interesting but could be further supported, please see my previous points 1 and 2. Figure 4C would benefit from additional controls, the native untagged protein treated identically). Also, please provide the corresponding full gel stained with Coomassie including the stacking gel. There are other bands detected by the antibody although less intense. Since this assay is done in the cell, the higher molecular weight bands could correspond to complexes with other thiol containing periplasmic proteins.

The MW of the proposed native dimer seems to be different to the dimers of the different mutants, which in that part of the SDS-PAGE gel the difference could be quite substantial. Please comment.

6. The statement "This finding suggests that F427 may be important for dimerization and provides a possible explanation for why its mutation to cysteine might hinder the formation of intermolecular crosslinks" may need some clarification. It is very likely that F427 is facing the hydrophobic membrane rather than the barrel lumen. If this residue facilitates dimerization via pi-pi interactions as I think it is suggested from the Alpha-fold model, the cysteine substitution should facilitate a disulphide bonded dimer unless the distance between the thiol groups is $> 3.0 \text{ \AA}$. That could be modelled on the alpha fold model (bearing in mind that this is a model based on another model).

7. Figure 4d. The authors should comment why there is so little dimer when oxidant is not added for all mutants. The periplasm is an oxidizing environment, as it is the extracellular environment so providing this is a native dimer, I would have expected to see more dimer without the need of external oxidizing conditions.

8. Figures 4d, e and f y axis should say inter-molecular disulphide

9. Figure 5 and statement "Surprisingly, when lid segment residues V113, I117 or S118 were converted to cysteine, a very high level of intermolecular disulphide bond formation was observed (~84%) (Fig. 5b, lanes 1-2, 21-22, and 25-26) showing that a PL263 domain lid can interact with its cognate lid segment ("lid-inter-lid"). In addition, when the lid mutations were combined with a mutation in β -strand 1 of the β -barrel (F410C, L411C, R412C, L413C or L415C) or the C-terminal turn of the β -barrel (L703C), the gel migration patterns not only indicated intra-molecular crosslinks ("lid-intra-barrel") (Fig. 5b, lanes 5-6, 9-10, 13-14, 17- 20, 23-24, and 29-30), but also inter-molecular crosslinks between the lid of one PlpD molecule and the barrel of the second PlpD molecule ("lid-inter-barrel") (Fig. 5b, lanes 14, 18, 20, and 29-30)."

The interpretation of these results need further clarification. It seems that the MW of the proposed

cross-linked dimers in Fig 5 is not the same as the one obtained in Fig 3. Also, the intensity of the reported "non specific" and some of the "specific" bands in not very different (i.e lanes 7,8 vs 9, 10 or lane 16 vs 17, 18). How was this conclusion reached? It is unclear why the different dimer forms of PlpD "lid-intra-barrel" and "lid-inter-barrel" would migrate differently on denaturing SDS-PAGE.

10. The authors state "The data not only confirms the predicted intramolecular interaction between the PL-domain lid with the β -barrel but also indicates that the lid slides dynamically across β -strand 1 of both β - barrels.

I am not sure whether the westerns show that this loop can interact with both β -barrels. Perhaps the authors can clarify further how they reach this conclusion.

11. For Fig. 5c and statement "Consistent with our hypothesis, the majority of PlpD molecules containing F102C and N513C substitutions in combination formed lid-intra-barrel disulphides".

Please clarify why the intramolecular disulphide bond leads to a different mobility in SDS-PAGE. Could the different mobility reflect the folded/unfolded state of the beta-barrel?

Fig S3, 4, and 6 and methods included in the methods section (Heat-dependent protein mobility shift assay, Protein purification and biophysical characterization) are not mentioned in the manuscript. Please correct.

Reviewer #3 (Remarks to the Author):

Hanson et al. present a manuscript examining the Omp85-phospholipase protein PlpD from *Pseudomonas aeruginosa*. Till now, this protein was considered the prototype of the type 5d secretion system, where a C-terminal Omp85 family β -barrel domain was thought to export the N-terminal phospholipase domain to the cell surface. The phospholipase was then suggested to be proteolytically cleaved from the translocator and released into the extracellular environment. Hanson et al. provide evidence for different topology for PlpD, where the phospholipase domain remains in the periplasm and is not cleaved from the β -barrel and POTRA domain. This topology is supported by an AlphaFold model. Furthermore, the authors provide evidence for dimerisation of the β -barrel domain and for a dynamic phospholipase 'lid', which interacts with the β -barrel domain and seems to alternate between an 'intra' and 'inter' β -barrel bound state.

The paper is well written and presented, and the biochemical work appears solid and corroborates the predictions based on the AlphaFold model. I only have a few minor comments on this below. The main finding in this paper is that "type 5d secretion" appears to be a misnomer, and that the phospholipase domains of PlpD and other Omp85-PLs are in fact retained in the periplasm. This is of interest to the type 5 secretion community and researchers working on Omp85 proteins. However, I am not sure whether this would be of general enough interest to the wider readership of Nature Communications, especially as the function of Omp85-PLs remains enigmatic. The authors speculate that they could be involved in lipid homeostasis of the outer membrane, but evidence for this is currently lacking. The dimerisation of the β -barrel domain, while interesting, is perhaps not very surprising given the known dimerisation of the phospholipase domains. Similarly, the dynamic lid structure is a very interesting observation, but its role in the function of the protein remains to be discovered.

Minor comments:

1. The topology of PlpD was probed only by proteolytic cleavage. While I tend to agree with the authors as to the conclusion, the proposed topology is in contrast to previous reports, so I would like

to see the topology confirmed by another method. The easiest one would seem to be performing fluorescence microscopy using the α -StrepII antibody on intact and permeabilised cells. It should be noted that surface exposure of the phospholipase was seen using this technique when the *Fusobacterium* protein FplA was expressed in *E. coli*.

4. The observation that the *P. aeruginosa* OM is permeable to proteases is surprising, particularly given that the OM of this organism is thought to be a particularly strong barrier. The authors do provide a control in the form of SurA, though the reduction of SurA levels is much less dramatic than the changes for PlpD. I would feel more confident about this procedure if a second control were to be included. This also supports the need for a second method to determine the topology.

3. Please provide more information about how the α -PlpPc antibody was generated - was this done in-house, or through a company? Presumably, a rabbit was immunised?

4. In Figure 3b, I would like to see a negative control, i.e. a pair of cysteines far apart in the structure that are not expected to interact. The authors provide something along these lines in Figure 4c (L359C), but something similar could be provided here, or the control in the other figure should already be alluded to.

5. In figures 4d-4f, shouldn't the Y axis be labelled 'inter' rather than 'intra'?

6. Do the authors have an explanation for why the interpolypeptide crosslinks in Fig. 4c are all well above 260 kDa, whereas the crosslinks in Fig. 5b are all below 260 kDa?

REPLY TO REVIEWERS' COMMENTS (all major changes in the text are highlighted)

Reviewer #1 (Remarks to the Author):

In this well-written manuscript, Hanson and colleagues report the investigation of *P. aeruginosa* OMP PlpD and provide new insights about its membrane orientation and quaternary structure. The results in this manuscript challenge the standing view that type Vd proteins: (1) are OM proteins that display /secrete the functional domain outside of the bacterial cell (similar to other Type V proteins), and (2) are monomeric proteins (similar to other OM beta barrel proteins).

Based on bacterial expression of PlpD wt and mutants, the latter designed informed by an alpha fold model of PlpD, combined with cross-linking experiments analyzed by immunoblotting, the authors propose that PlpD is homodimeric with the PL-domains located in the periplasm.

This is really exciting and although the experiments are logical, given the implications of the proposed paradigm change for PlpD specifically and Type Vd proteins in general, this work would benefit from additional supporting data to support the claims.

We would like to thank the reviewer for his/her positive remarks.

Specific comments include the following:

1. The long N-terminal signal sequence is key for the transportation and folding of all Type5 proteins. I worry about the addition of the TwinStrepII (TS) tag (28 aminoacids) at the N-terminus of the PL-domain. This could alter the processing and translocation of the protein across the OM. Was the experiment conducted with the un-tagged protein and detected with the antiserum generated against a C-terminal PlpD peptide (aPlpDC) to see whether full length or processed protein (without the PL domain) was present in the membrane fraction?

As suggested by the reviewer, we repeated the experiments shown in Fig. 1c-d with an untagged version of PlpD and obtained essentially the same results we obtained when we analyzed the TS-tagged version. In both cases PK treatment did not generate a C-terminal fragment that corresponds to the linker, POTRA domain and barrel (which would have been expected if the protease removed a surface-exposed PL-domain), and the entirety of PlpD was resistant to chymotrypsin cleavage. The only difference is that while both proteases reduced the size of the tagged version of PlpD by ~2-3 kDa by removing the tag, they left the untagged version of the protein completely intact. The new data are now shown in Fig. S3 and the text has been modified accordingly (lines 157-159 and 170-172). Furthermore, to provide additional evidence that the introduction of the tag has no effect on the assembly of PlpD, we now also show in that both tagged and untagged forms of the protein exhibit the same folding properties in a heat-dependent mobility shift assay (Fig. S2; lines 128-131).

We should also note that neither PlpD nor other members of the PL-Omp85 family have extended (~50 residue) signal peptides. Long signal peptides appear to be restricted to a subset of classical (Type Va) autotransporters and members of the two-partner secretion (Type Vb) family.

2. It would be important to show that the purified recombinant proteins carrying the TS tag, are fully folded. Some data is presented in supplementary figures 4, which is not discussed or mentioned in the manuscript.

Figure 4S data is crucial to support the rest of the manuscript and the authors could add some missing information

We refer to Fig. S4 (now Fig. S6) in the text (lines 186-189), but we do not describe the experiments in detail in the interest of focusing on the more important aspects of our study. The experiments shown in Figs. S5-S6 were conducted in *E. coli* only to obtain some basic information about PlpD, and while they support our main conclusions, our goal was to focus on the structure of PlpD in its native host [See reply to comment 2(ii)].

(i) The Coomassie stained SDS-PAGE gel of the purified proteins corresponding to the samples used for biophysical characterisation should be included. To confirm that the beta-barrel is folded in these constructs I would suggest including a gel with samples both, boiled and not boiled. The Heat-dependent protein mobility shift assay is mentioned in the methods section but no related data is shown in the manuscript.

We performed the heat-dependent mobility shift assay suggested by the reviewer and found that the purified proteins are indeed correctly folded. The results are now shown in Fig. S6c. It is also important to note that the observation that the purified proteins are completely resistant to proteolysis (except for the unstructured tag; Fig. 6a) further supports our conclusion that they are correctly folded.

(ii) Ideally the recombinant production and purification of the full length PlpD should be presented, given that this is the protein that is proposed to have a different 3D structure and oligomeric state to other type V proteins. The secondary structure checked by CD, apparent MW determined by biophysical techniques such as DLS, analytical ultracentrifugation, size exclusion chromatography (perhaps using one of the deletion mutants (i.e. monomeric LPB) as control). If not enough full length recombinant protein can be obtained for purification, the authors can consider analyzing the native and all relevant mutants using heat-dependent protein mobility shift assay and native PAGE gels.

The reviewer is referring to experiments in *E. coli* that were performed only at the beginning of our study to obtain some basic information about the topology of the protein. We found that the full-length protein is completely resistant to PK digestion in intact *E. coli* cells (Fig. S5a). When the cells are first permeabilized, however, some of the protein is cleaved, but the POTRA-barrel fragment remains intact. These results provided the first indication that the PL domain is retained in the periplasm, the POTRA-barrel fragment is properly folded, and the barrel is inserted into the OM. We subsequently repeated the PK digestion experiment with truncated forms of PlpD to obtain further evidence that the POTRA domain is properly folded and that the barrel is inserted into the OM even without the PL domain (Fig. S5b). The biochemical and biophysical experiments shown in Fig. S6 were performed primarily to bolster the data shown in Fig. S5b by demonstrating that the POTRA domain folds correctly even when truncated forms of PlpD are purified and, perhaps more importantly, that the barrel is correctly folded. After obtaining these results we really wanted to focus on the fate of PlpD in its native host, *P. aeruginosa*, which is the subject that we present in the main manuscript. The primary reason that we included the *E. coli* data in the Supplement is that in a heat-dependent mobility shift assay we obtained the first evidence that the C-terminal domains of PlpD form a dimer (Fig. S5c). As mentioned in the Introduction, the N-terminal PL-domain had already been shown to be a dimer by x-ray crystallography. At this point, we are uncertain what we will gain by purifying the full-length protein from *E. coli* and characterizing it.

3. Fig 1b, The authors indicate that they detect a band corresponding to full-length PlpD (~82

kDa) in cell pellet samples. Could this indicate that the protein is not correctly folded and therefore not translocated and processed.

The reviewer raises a good point. The PK data in Fig. 1c and the chymotrypsin data in Fig. 1d, however, show that full-length protein is localized to the OM and properly folded. The TS tag was particularly useful in Fig. 1d because it was cleaved off the full-length protein while the entire protein remained intact (see lanes 3-4). That observation provides strong evidence that PlpD was correctly folded. The new data shown in Fig. S2 indicate that full-length PlpD is resistant to unfolding by SDS and therefore displays an intrinsic feature of OMPs that are correctly assembled. Finally, the highly specific nature of the *in vivo* crosslinking data presented in Figs. 3, 4, and 5 very strongly suggests that the protein is properly folded.

Fig 1c. The authors state that in *Pseudomonas* the protease is able to permeabilise the OM. in this case the +/- permeabilization experiment seems redundant since it cannot inform about whether the PL domain has been translocated. These experiments show that the full-length protein including the TS tag is present in the pellet which could be because the TS tag impedes correct translocation outside of the cell. I suggest carrying out the experiment with an untagged construct or not tagged at the n-terminus.

We performed the experiment suggested by the reviewer and also repeated the experiment shown in Fig. 1d. We found that we obtained the same results when we treated an untagged form of the protein with PK or chymotrypsin. The results are shown in Fig. S3 and we have modified the text accordingly (see lines 157-159 and 170-172).

4. Figure 3. It is unclear why the formation of an intramolecular disulphide bond upon mutation of the predicted interacting residues into C and thiol oxidation with 4-DPS would lead to increasing the MW on SDS-PAGE.

What is the apparent MW of the new band? Were these samples also run in the presence of reducing agent and did the higher MW band disappear. I would also suggest testing the un-boiled proteins in parallel. Were the single mutants in the Potra domain Q383C and V388C tested? Could the experiment be trapping a complex between PlpD and another periplasmic proteins?

Intramolecular disulfide bonding typically alters the mobility of proteins by changing their structure, but it is often not possible to predict whether the disulfide bond will increase or decrease mobility. As we now note in the text, intramolecular disulfide bonding has previously been observed to reduce the mobility of other proteins, including a member of the Omp85 family called Sam50 (see lines 213-214 and ref. 48).

The proteins that would most likely interact with PlpD during its assembly are chaperones such as Skp, SurA, FkpA and DegP (all of which are produced by *P. aeruginosa*), but none of them contain any cysteine residues and therefore could not form a disulfide bond with PlpD. In our experience, at least in *E. coli* DsbA does not typically form disulfide bonds between cysteine residues unless they come close together as a protein folds. As we showed in ref. 22 and Wang and Bernstein (2022) JBC 298:101802, DsbA will not form disulfide bonds between cysteine residues that are only slightly out of register.

It is important to note that we never see the higher molecular weight bands when PlpD contains only one cysteine residue—we only see them when the protein contains two cysteines. In any case, to address the reviewer's concern and to support the notion that the higher molecular weight bands correspond to intramolecular crosslinking products, we re-ran the samples in the

presence of a reducing agent and found that they disappeared. The new results are now shown in Fig. S8 and the text has been modified accordingly (lines 214-217).

5. The dimerization of PlpD is very interesting but could be further supported, please see my previous points 1 and 2. Figure 4C would benefit from additional controls, the native untagged protein treated identically). Also, please provide the corresponding full gel stained with Coomassie including the stacking gel. There are other bands detected by the antibody although less intense. Since this assay is done in the cell, the higher molecular weight bands could correspond to complexes with other thiol containing periplasmic proteins.

The MW of the proposed native dimer seems to be different to the dimers of the different mutants, which in that part of the SDS-PAGE gel the difference could be quite substantial. Please comment.

To address the reviewer's concern, we now show in Fig. S11 that the untagged version of PlpD forms the same pattern of intermolecular disulfide bonds as TS-PlpD and we have modified the text accordingly (lines 250-252).

It should be noted that in Fig. 4C we show Western blots and not Coomassie stained gels. The less intense bands are very unlikely to represent crosslinks between PlpD and other periplasmic proteins because there are few, if any, periplasmic proteins in the 80-100 kD size range. It is much more likely that these bands are partial breakdown products or forms of the crosslinked dimer that migrate differently on the gel. In this regard we should note that intermolecular crosslinking products between different residues in the same proteins have been previously observed to migrate a bit differently on SDS-PAGE (as we now note on lines 248-250 along with two new references). Intermolecular disulfide adducts formed between cysteine residues nearer to protein termini are expected to migrate differently than adducts of the same proteins crosslinked near their mid-points simply because the conformations of the adducts are different (as observed in ref. 51).

6. The statement "This finding suggests that F427 may be important for dimerization and provides a possible explanation for why its mutation to cysteine might hinder the formation of intermolecular crosslinks" may need some clarification. It is very likely that F427 is facing the hydrophobic membrane rather than the barrel lumen. If this residue facilitates dimerization via pi-pi interactions as I think it is suggested from the Alpha-fold model, the cysteine substitution should facilitate a disulphide bonded dimer unless the distance between the thiol groups is > 3.0 Å. That could be modelled on the alpha fold model (bearing in mind that this is a model based on another model).

If the binding between the two β -barrel domains is stable (as we suspect), then the orientation of the introduced cysteine residues is crucial. They would need to be in a specific orientation to form a disulfide bond. According to the Alphafold prediction, the two F427 residues are rotated away from each other. Indeed if cysteine residues introduced at position 427 adopt the same rotamer, they may not be able to form a disulfide bond efficiently. As we state in the text, we do see some disulfide bond formation, so F427 (which is located at the dimer interface) may indeed face inward to the barrel lumen and therefore not be readily available to form an intermolecular disulfide bond. Alternatively, because it is a conserved residue, mutating it to cysteine might alter the conformation of the protein slightly and thereby hinder disulfide bond formation. We have now modified the text slightly to clarify this last point (lines 257-258).

It is well established that AlphaFold does not work well to predict the effect of point mutations on protein structure, so we are just providing possible explanations of the data. As an aside, if PlpD is a phosphatase that captures lipids near the dimer interface (as we suggest in the last paragraph of the Discussion and Fig. S10), then it seems reasonable that a few large hydrophobic residues point inward.

7. Figure 4d. The authors should comment why there is so little dimer when oxidant is not added for all mutants. The periplasm is an oxidizing environment, as it is the extracellular environment so providing this is a native dimer, I would have expected to see more dimer without the need of external oxidizing conditions.

The reviewer raises a good point here. We and others have repeatedly observed low levels of disulfide bond formation *in vivo* in the absence of an oxidizing agent (see refs. 22-24 and, for example, Noinaj N et al. (2014) Structure 22: 1055-1062). Likewise, low levels of disulfide bond formation are observed in yeast mitochondria in the absence of an oxidizing agent (see, for example, ref. 48), which also have an oxidizing intermembrane space (the equivalent of the bacterial periplasm). We also know that naturally occurring disulfide bonds do not form in *E. coli* in the absence of DsbA. Clearly the oxidizing environment of the periplasm is not yet fully understood.

8. Figures 4d, e and f y axis should say inter-molecular disulphide

We have corrected this error.

9. Figure 5 and statement “Surprisingly, when lid segment residues V113, I117 or S118 were converted to cysteine, a very high level of intermolecular disulphide bond formation was observed (~84%) (Fig. 5b, lanes 1-2, 21-22, and 25-26) showing that a PL263 domain lid can interact with its cognate lid segment (“lid-inter-lid”). In addition, when the lid mutations were combined with a mutation in β -strand 1 of the β -barrel (F410C, L411C, R412C, L413C or L415C) or the C-terminal turn of the β -barrel (L703C), the gel migration patterns not only indicated intra-molecular crosslinks (“lid-intra-barrel”) (Fig. 5b, lanes 5-6, 9-10, 13-14, 17- 20, 23-24, and 29-30), but also inter-molecular crosslinks between the lid of one PlpD molecule and the barrel of the second PlpD molecule (“lid-inter-barrel”) (Fig. 5b, lanes 14, 18, 20, and 29-30).”

The interpretation of these results need further clarification. It seems that the MW of the proposed cross-linked dimers in Fig 5 is not the same as the one obtained in Fig 3. Also, the intensity of the reported “non specific” and some of the “specific” bands in not very different (i.e lanes 7,8 vs 9, 10 or lane 16 vs 17, 18). How was this conclusion reached? It is unclear why the different dimer forms of PlpD “lid-intra-barrel” and “lid-inter-barrel” would migrate differently on denaturing SDS-PAGE.

We identified non-specific crosslinks based on the data obtained using single cysteine substitutions located in the first strand of the PlpD β -barrel (410, 412, 413, 415). Because these residues are not expected to interact with the cognate residues in the adjacent barrel when the dimer is fully folded in the OM, the faint bands that we label as “non-specific” likely arose from fortuitous interactions of nascent PlpD molecules in the periplasm prior to their assembly into the OM. Similar background bands are commonly seen in this type of experiment (see, for example, refs. 22 and 51). We have now modified the text to clarify this point (lines 280-282).

To address the reviewer’s concern, we should note that the intramolecular crosslinking products shown in Fig. 3 would be expected to be much smaller than the intermolecular crosslinking

products we see in Fig. 5 because the latter products contain two >80 kD subunits. As mentioned above, crosslinking products often migrate very differently on SDS-PAGE than their expected molecular weight, and it is often difficult to predict their mobility a priori. The lid-inter-barrel products migrate differently than the lid-inter-lid products because the covalent attachment of different segments of the two subunits significantly affects mobility. The intrabarrel crosslinking products observed in Fig. 5b-c migrate in the ~100 kDa range, just like the intrabarrel crosslinking products observed in Fig. 3. As in Fig. 3, the intrabarrel crosslinking products in Fig. 5 migrate slightly differently from each other because different residues are involved in disulfide bond formation.

10. The authors state “The data not only confirms the predicted intramolecular interaction between the PL-domain lid with the β -barrel but also indicates that the lid slides dynamically across β -strand 1 of both β -barrels.

I am not sure whether the westerns show that this loop can interact with both β -barrels. Perhaps the authors can clarify further how they reach this conclusion.

Our conclusion is based on the observation that residues in the lid domain (e.g., 113, 117) form both intramolecular and intermolecular disulfide bonds with residues in the β -barrel domain (e.g., 411, 413, 415, 703). See, for example, Fig. 5b, lanes 14 and 20. Our explanation of the data appears in the preceding sentence (lines 285-291). Importantly, in the controls shown in 5c, no lid-inter-barrel crosslinks are observed as the residues are on the opposite sides of the complex.

11. For Fig. 5c and statement “Consistent with our hypothesis, the majority of PlpD molecules containing F102C and N513C substitutions in combination formed lid-intra-barrel disulphides”.

Please clarify why the intramolecular disulphide bond leads to a different mobility in SDS-PAGE. Could the different mobility reflect the folded/unfolded state of the beta-barrel?

As stated above (see reply to questions 4, 5 and 9), it has been shown previously that the formation of disulfide bonds between different residues within a protein or between two different polypeptides can alter mobilities on SDS-PAGE depending on where the crosslinks are situated. The samples were all heated before SDS-PAGE, so the β -barrels were likely fully unfolded.

Fig S3, 4, and 6 and methods included in the methods section (Heat-dependent protein mobility shift assay, Protein purification and biophysical characterization) are not mentioned in the manuscript. Please correct.

The methods that the reviewer refers to were used in experiments shown in Supplementary Figures, but not in the experiments shown in the main Figures. We could have placed the details of the heat-dependent protein mobility shift assay, protein purification and biophysical characterization in a Supplementary Methods section, but we believe that it is preferable to consolidate all of the methods in one place.

Reviewer #3 (Remarks to the Author):

Hanson et al. present a manuscript examining the Omp85-phospholipase protein PlpD from *Pseudomonas aeruginosa*. Till now, this protein was considered the prototype of the type 5d secretion system, where a C-terminal Omp85 family β -barrel domain was thought to export the

N-terminal phospholipase domain to the cell surface. The phospholipase was then suggested to be proteolytically cleaved from the translocator and released into the extracellular environment. Hanson et al. provide evidence for different topology for PlpD, where the phospholipase domain remains in the periplasm and is not cleaved from the β -barrel and POTRA domain. This topology is supported by an AlphaFold model. Furthermore, the authors provide evidence for dimerisation of the β -barrel domain and for a dynamic phospholipase 'lid', which interacts with the β -barrel domain and seems to alternate between an 'intra' and 'inter' β -barrel bound state.

The paper is well written and presented, and the biochemical work appears solid and corroborates the predictions based on the AlphaFold model. I only have a few minor comments on this below. The main finding in this paper is that "type 5d secretion" appears to be a misnomer, and that the phospholipase domains of PlpD and other Omp85-PLs are in fact retained in the periplasm. This is of interest to the type 5 secretion community and researchers working on Omp85 proteins. However, I am not sure whether this would be of general enough interest to the wider readership of Nature Communications, especially as the function of Omp85-PLs remains enigmatic. The authors speculate that they could be involved in lipid homeostasis of the outer membrane, but evidence for this is currently lacking. The dimerisation of the β -barrel domain, while interesting, is perhaps not very surprising given the known dimerisation of the phospholipase domains. Similarly, the dynamic lid structure is a very interesting observation, but its role in the function of the protein remains to be discovered.

We thank the reviewer for his/her positive remarks. We have now included some completely new data that addresses the function of PlpD that we believe will increase the interest of our manuscript to a wide readership (see reply to comment 1).

Minor comments:

1. The topology of PlpD was probed only by proteolytic cleavage. While I tend to agree with the authors as to the conclusion, the proposed topology is in contrast to previous reports, so I would like to see the topology confirmed by another method. The easiest one would seem to be performing fluorescence microscopy using the α -StreptII antibody on intact and permeabilised cells. It should be noted that surface exposure of the phospholipase was seen using this technique when the *Fusobacterium* protein FplA was expressed in *E. coli*.

The reviewer refers to ref. 44 in which the authors used an antiserum raised against amino acids 20-431 of FplA for the immunofluorescence experiments. The amino acid sequence corresponds not only to the PL-domain, but also to the POTRA domain, and the antiserum could recognize epitopes located in only one of the domains. Furthermore, it should be noted that the cells appeared to be undergoing lysis, which very possibly allowed the antiserum to access the periplasm. Finally, the experiments were conducted in *E. coli*, an organism that is very distantly related to *Fusobacterium*. These points all raise concerns regarding the interpretation of the results.

In any case, to address the reviewer's concern we have put a great deal of effort into confirming the topology of PlpD in its native host (*P. aeruginosa*) using a second method. We have tried several different "standard techniques" including, dot blotting, and pegylation, but none of these methods provided clearly interpretable results. We were extremely fortunate, however, that a completely different experimental approach both confirmed the topology of the protein and yielded some interesting insights into its function. Through a collaboration with our NIH colleague Dr. Mioara Larion, an authority on lipidomics, and her associate, Tyrone Dowdy, we found that the phospholipid profiles of *P. aeruginosa* strain PA14 (the wild-type strain) and an isogenic $\Delta plpD$ strain are dramatically different. In short, we found that the levels of a wide

variety of phospholipids are significantly elevated in the mutant strain. The results strongly suggest that, as predicted from its sequence and structure, PlpD is a phospholipase. Moreover, the results also confirm the topology of the PL-domain and provide very strong support for our model in which the lipid hydrolyzing pocket is specifically positioned beneath the inner leaflet of the OM. Because phospholipids are excluded from the outer leaflet of the OM (as recently shown in ref. 58), it would be difficult to imagine how an enzyme that cleaves phospholipids that are confined to the inner leaflet of the OM would be located on the cell surface. The new lipidomic data are described and discussed on lines 318-336, 401-412, and 570-646 (Methods section) and shown in Fig. 6, Fig. S13, and Table S1, and of course Dr. Larion and Mr. Dowdy and now included as co-authors.

2. The observation that the *P. aeruginosa* OM is permeable to proteases is surprising, particularly given that the OM of this organism is thought to be a particularly strong barrier. The authors do provide a control in the form of SurA, though the reduction of SurA levels is much less dramatic than the changes for PlpD. I would feel more confident about this procedure if a second control were to be included. This also supports the need for a second method to determine the topology.

The reviewer raises a good point here. In response to his/her concern we have looked at the sensitivity of other periplasmic proteins to proteases by using antisera raised against *E. coli* proteins that are conserved in *P. aeruginosa*, but the results are ambiguous. The lipid-hydrolyzing activity of PlpD itself (which was confirmed by our lipidomic analysis) might be responsible for the permeability nuances that we observed. It seems possible that the OM is not fully permeable, but that PK has access to OMPs and factors like SurA, which binds to the Bam complex and is therefore not only close to the OM but also associated with a factor that is known to perturb the local lipid environment during its functional cycle. To clarify the results, we have added the words "a large fraction of" before "the periplasmic chaperone SurA was also cleaved by the proteases" on line 184.

3. Please provide more information about how the α -PlpPc antibody was generated - was this done in-house, or through a company? Presumably, a rabbit was immunised?

We have provided the requested information (see lines 509-510).

4. In Figure 3b, I would like to see a negative control, i.e. a pair of cysteines far apart in the structure that are not expected to interact. The authors provide something along these lines in Figure 4c (L359C), but something similar could be provided here, or the control in the other figure should already be alluded to.

We believe that the single cysteine mutations in Fig. 3b are good negative controls because they do not show any intramolecular or intermolecular crosslinking. The experiment shown in Fig. 4 is somewhat different from the experiment shown in Fig. 3, and it might be confusing to allude to the L359C results shown in Fig. 4 when we discuss the results shown in Fig. 3. To address the reviewers concern we now show in Fig. S11 (lanes 1-2) that we only observe background levels of non-specific crosslinking between distantly located residues.

5. In figures 4d-4f, shouldn't the Y axis be labelled 'inter' rather than 'intra'?

We have corrected this error.

6. Do the authors have an explanation for why the interpolypeptide crosslinks in Fig. 4c are all well above 260 kDa, whereas the crosslinks in Fig. 5b are all below 260 kDa?

Actually, the lid-inter-barrel crosslinks in Fig. 5b (see, for example, lanes 16 and 18) are also above 260 kDa. In any case, as we mention in our reply to reviewer 1, both intramolecular and intermolecular crosslinks often do not migrate on SDS-PAGE at their expected molecular weight (see reply to questions 4, 5 and 9). As we and other investigators have previously observed, in some cases two proteins that are crosslinked to each other at different residues migrate differently presumably because the conformation of the two covalently linked proteins diverge and therefore affect the mobility of the complex differently. We have now cited two additional references and modified the text (lines 248-250) to address the general issue of the differential mobility of crosslinked proteins that are joined together at different positions.

Reviewer #3 (Remarks to the Author):

The authors have done a good job of responding to my comments. The lipidomics analysis offers a tantalising hint as to the function of PlpD, and this is an excellent addition to the manuscript. The only minor comment I would have on this is that Figure 6B might be improved, as it is not clear which dot the labels refer to, and one of the labels is obscured by the dotted line. Also, what does MG stand for? This is a very minor quibble, however.

Reviewer #4 (Remarks to the Author):

The authors have performed a lipidomic analysis of the WT and the Δ plpD strains of *P. aeruginosa* in an attempt to demonstrate the reported phospholipase activity of PlpD. The data processing and identification of lipid species from this experiment are not to level of rigor that is expected for LC-MS based lipid analyses. The authors should revise the interpretation of the data to be consistent with the level of details that is appropriate based on their experimental methods. The Lipidomics Minimal Reporting Checklist developed by the Lipidomics Standards Initiative will be a useful guide for the authors in revising their lipidomics identifications.

1. The lipidomics data are substantially over-annotated based on the methods used in this study. It is unacceptable to provide lipid identifications to the acyl tail positions (e.g., PA 16:0/18:1) using accurate mass alone. Tandem MS (MS/MS) is required to provide identifications to this level. With accurate mass, the best that can be done is "total carbon:total unsaturation" nomenclature (e.g., PA 34:1), and even that leaves a lot of uncertainty. An additional table should be provided as SI indicating retention time, the m/z, the adduct (e.g., [M-H]⁻ or [M+Na]⁺) and mass error for each putative identification. If there is more than one possible ID, it should be listed in this table as well.
2. It is suspicious that the volcano plot shows such a strong bias towards the Δ plpD mutant. I suspect that the normalization method may be contributing to this. Why normalize to protein concentrations? Were the amounts of protein between the two strains very different? From the methods, it appears that attempts were made to use similar numbers of cells for the extraction. If equivalent cell densities (CFU/mL) were extracted, what is the purpose of normalizing the data to protein concentration?
3. Generally, I don't think the lipid data supports the authors' claim that PlpD must be in the periplasm just because many lipids are changed upon knocking out the gene. It's a very broad generalization and many other factor may be induced upon a KO that could change lipid biosynthesis.
4. The authors indicate in the discussion that PlpD has activity similar to phospholipase A1, but that lipase selectively cleaves the acyl tail at the sn-1 position of phospholipids for lipid remodeling. I'd expect to see a decrease in lyso-phospholipids and elevated phospholipids. SI Table 1 does show that LysoPE 16:0 and LysoPG 18:1 are decreased in the mutant with decent fold-changes (0.55 and 0.65). Those may be worth investigating more or highlighting as the authors revise their interpretation of the lipid analysis.
5. The authors claim that this is "quantitative" but I see no evidence that appropriate stable isotope labeled lipid internal standards were used, nor that quantitation is provided in the manuscript. phenyl-N-pyridinyl acrylamide is not an appropriate internal standard for lipids. A comparative experiment (no quantitation, just relative differences) is fine for this type of study.
6. Data has been collected in positive and negative ionization modes. It is unclear from the methods/results how that data was used. Were detected features merges between data from the two ionization modes? Was only one data set used? How did the authors "de-replicate" putative lipid IDs when the chromatographic methods for the two experiments were different?

7. The SI table lists oleate and vaccenic acid. These are both C18:1 fatty acids that differ only in the location of the double bond ($\Delta 9$ vs $\Delta 11$). I doubt that these have been resolved by the chromatography.

8. Based on the use of the Bligh & Dyer extraction, it is unreasonable to identify small, polar metabolites such as keto isovaleric acid, urea, etc.

REPLY TO REVIEWERS' COMMENTS (all significant changes are highlighted)

Reviewer #3 (Remarks to the Author):

The authors have done a good job of responding to my comments. The lipidomics analysis offers a tantalising hint as to the function of PlpD, and this is an excellent addition to the manuscript. The only minor comment I would have on this is that Figure 6B might be improved, as it is not clear which dot the labels refer to, and one of the labels is obscured by the dotted line. Also, what does MG stand for? This is a very minor quibble, however.

To address this concern we have now spelled out the names of all the classes of lipids that we abbreviated in Fig. 6B (e.g., we now state that “MG” stands for monoacylglycerol).

Reviewer #4 (Remarks to the Author):

The authors have performed a lipidomic analysis of the WT and the Δ plpD strains of *P. aeruginosa* in an attempt to demonstrate the reported phospholipase activity of PlpD. The data processing and identification of lipid species from this experiment are not to level of rigor that is expected for LC-MS based lipid analyses.

The authors should revise the interpretation of the data to be consistent with the level of details that is appropriate based on their experimental methods. The Lipidomics Minimal Reporting Checklist developed by the Lipidomics Standards Initiative will be a useful guide for the authors in revising their lipidomics identifications.

We thank the reviewer for his/her helpful comments.

1. The lipidomics data are substantially over-annotated based on the methods used in this study. It is unacceptable to provide lipid identifications to the acyl tail positions (e.g., PA 16:0/18:1) using accurate mass alone. Tandem MS (MS/MS) is required to provide identifications to this level. With accurate mass, the best that can be done is “total carbon:total unsaturation” nomenclature (e.g., PA 34:1), and even that leaves a lot of uncertainty. An additional table should be provided as SI indicating retention time, the m/z, the adduct (e.g., [M-H]⁻ or [M+Na]⁺) and mass error for each putative identification. If there is more than one possible ID, it should be listed in this table as well.

To address this concern, we now use a standardized “total carbon: total unsaturation” nomenclature that should be acceptable for accurate mass-based putative lipid identifications. In addition, a new Table has been added to the Supplement (Table S10) that shows the m/z along with the corresponding adduct (e.g., [M-H]⁻ or [M+H]⁺), the retention time, and the mass error for each putative assignment. A column has been included in the Table to show other possible IDs. It should be noted, however, that as we describe in the Methods section, the putative assignments are based on mass accuracy, similar elution of other previously validated lipids from the same lipid class, and the Meta Cyc and PAMD confirmation of features that were previously identified in the *Pseudomonas* lipidome.

2. It is suspicious that the volcano plot shows such a strong bias towards the Δ plpD mutant. I suspect that the normalization method may be contributing to this. Why normalize to protein concentrations? Were the amounts of protein between the two strains very different? From the methods, it appears that attempts were made to use similar numbers of cells for the extraction.

If equivalent cell densities (CFU/mL) were extracted, what is the purpose of normalizing the data to protein concentration?

We follow the common practices described in The Lipidomics Minimal Reporting Checklist developed by the Lipidomics Standards Initiative (<https://lipidomicstandards.org/lipid-species-quantification/>) and, in our experience, protein normalization is much more accurate than cell densities because errors arise when cells are counted and cells can be lost during the collection process. As described in the Method section, all standard practices (randomized sample extraction, randomized run order, equal distribution of qualitative internal standard, normalization to sample-specific protein concentration as well as sample group analysis performed without knowledge of which group number was assigned to specific group classifications until the completion/reporting of group comparisons) were taken to avoid the introduction of bias.

Were the amounts of protein between the two strains very different?

As in typical lipidomic/metabolomic analyses that use cells, the protein concentration in each strain was slightly different. The wild-type strain had a group mean \pm SD of 12.9 ± 2.6 mg whereas the PlpD knockout strain had a \pm SD of 8.7 ± 1.1 mg.

3. Generally, I don't think the lipid data supports the authors' claim that PlpD must be in the periplasm just because many lipids are changed upon knocking out the gene. It's a very broad generalization and many other factors may be induced upon a KO that could change lipid biosynthesis.

We agree with the reviewer that the lipid data alone do not show that the PL-domain is located in the periplasm. While the deletion of any gene might have many physiological effects, however, it is difficult to imagine how the deletion of a gene that encodes an outer membrane phospholipase would affect phospholipid levels if the phospholipid domain is located on the cell surface or in the extracellular environment. The outer leaflet of the outer membrane is composed of LPS, and there are no substrates on the cell surface for the enzyme to act on.

Nevertheless, in light of the reviewer's comment, we have tried to interpret the data more cautiously. After all, because all of the lipid biosynthetic pathways are linked it is difficult to know if some of the changes in lipid levels are simply secondary effects of changes that result from the cleavage of a subset of phospholipids by PlpD. In the paragraph that begins on line 396 we have changed "The lipid differential that we discovered indicates that the PL-domain of PlpD cleaves a variety of phospholipids" to "The lipid differential that we discovered strongly suggests that the PL-domain of PlpD cleaves multiple phospholipids". We already noted in the previous version of the manuscript that "we cannot determine if the elevation of the levels of some of the lipids in the $\Delta plpD$ strain results from a cellular response to the loss of PlpD" but we have now changed this statement to "we cannot determine if the elevation of the levels of some of the lipids in the $\Delta plpD$ strain is a secondary effect of a cellular response to the loss of PlpD". Furthermore, we now provide an example of such a secondary effect by stating that "under normal conditions bacterial cardiolipin synthase uses PG as a substrate to produce CL, so an increase in PG levels in a $\Delta plpD$ strain might lead to an increase in CL." We have also changed "Because phospholipids are largely excluded from the outer leaflet of the OM in *Pseudomonas* it stands to reason that the PlpD PL-domain would have to reside in the periplasm to cleave endogenous phospholipids" to "Because phospholipids are largely excluded from the outer leaflet of the *Pseudomonas* OM it is very likely that the PlpD PL-domain would have to reside in the periplasm to cleave endogenous phospholipids." Finally, we end the paragraph by citing a paper that describes a bacterial patatin-like phospholipase that functions intracellularly. We

believe that by showing that related proteins have previously been shown to function intracellularly will bolster our conclusion that PlpD functions in the periplasm.

4. The authors indicate in the discussion that PlpD has activity similar to phospholipase A1, but that lipase selectively cleaves the acyl tail at the sn-1 position of phospholipids for lipid remodeling. I'd expect to see a decrease in lyso-phospholipids and elevated phospholipids. SI Table 1 does show that LysoPE 16:0 and LysoPG 18:1 are decreased in the mutant with decent fold-changes (0.55 and 0.65). Those may be worth investigating more or highlighting as the authors revise their interpretation of the lipid analysis.

We agree with the reviewer that it would be worth investigating these observations further in the future. Given that our main goal was to determine the structure and topology of PlpD and that a great deal of work would be required to distinguish between primary and secondary effects of the $\Delta plpD$ mutation, we believe that further analysis of the lipidomic data is beyond the scope of the present study.

5. The authors claim that this is "quantitative" but I see no evidence that appropriate stable isotope labeled lipid internal standards were used, nor that quantitation is provided in the manuscript. phenyl-N-pyridinyl acrylamide is not an appropriate internal standard for lipids. A comparative experiment (no quantitation, just relative differences) is fine for this type of study.

We agree with the reviewer, and in the interest of accuracy have now changed "quantitative" to "comparative" (see line 321) or referred to "relative levels" (see line 397) and have removed the word "quantitative" from the legend to Fig. 6.

6. Data has been collected in positive and negative ionization modes. It is unclear from the methods/results how that data was used. Were detected features merged between data from the two ionization modes? Was only one data set used? How did the authors "de-replicate" putative lipid IDs when the chromatographic methods for the two experiments were different?

The Method section has been updated to specify that the positive and negative ionization mode data were merged to conduct multivariate analysis and examine correlations in heatmap and relevance volcano plots (see lines 636, 639-640, and 650-651). Now that we have adopted a standard nomenclature, replication of putative assignments is more evident and features with repeated m/z are now labeled as "iso" or "iso 1,2,3...." where there are multiple features within the same mode. While there are fewer lipidomic features detected in negative mode, features from different ionization modes with the same m/z are now labeled as "iso p" for ESI+ and "iso n" ESI-.

7. The SI table lists oleate and vaccenic acid. These are both C18:1 fatty acids that differ only in the location of the double bond ($\Delta 9$ vs $\Delta 11$). I doubt that these have been resolved by the chromatography.

We have previously performed studies in which we validated a panel of fatty acids and lipids using MS/MS analysis to confirm specific retention times to develop a PCDL library of targets specific to the positive and negative mode with IDs with specific peak retention times and neutral mass for global analysis. In some cases, our high-resolution Q-TOF and high-pressure LC column methodology was sufficient to resolve isomers. As we mentioned in our reply to comment 1, however, despite differences in retention time we have changed our nomenclature and labeled repeated features as isomers in Table S10 to improve the scientific rigor of a lipidomic analysis in which we did not perform MS/MS analysis at the time of data acquisition.

8. Based on the use of the Bligh & Dyer extraction, it is unreasonable to identify small, polar metabolites such as keto isovaleric acid, urea, etc.

We initially decided to disclose all putative assignments including polar metabolites that are the only potential IDs based on accurately matching observed masses to masses listed in available databases for *Pseudomonas aeruginosa* (PMDB and EcoCyc). In light of the reviewer's comment, however, these polar metabolites have been removed from the reporting table.

Reviewer #4 (Remarks to the Author):

The authors have thoroughly revised the lipidomic results to be inline with the standards for the field.